# An Improved Federated Clustering Algorithm with Model-based Clustering

**Harsh Vardhan** *hharshvardhan@ucsd.edu*
*Halicioğlu Data Science Institute*
*University of California, San Diego*

**Avishek Ghosh** *avishek_ghosh@iitb.ac.in*
*Systems and Control Engg.,*
*Centre for Machine Intelligence and Data Sciences*
*Indian Institute of Technology, Bombay*

**Arya Mazumdar** *arya@ucsd.edu*
*Halicioğlu Data Science Institute*
*University of California, San Diego*

**Reviewed on OpenReview:** *https://openreview.net/forum?id=1ZGA5mSkoB*

## Abstract

Federated learning (FL) is a distributed learning paradigm that allows multiple clients to collaboratively train a shared model via communications to a central server. However, optimal models of different clients often differ due to heterogeneity of data across clients. In this paper, we address the dichotomy between heterogeneous models and simultaneous training in FL via a clustering structure among the clients. The clustering framework is one way to allow for high heterogeneity level between clients, while clients with similar data can still train a shared model. We define a new clustering framework for FL based on the (optimal) local models of the clients: two clients belong to the same cluster if their local models are close. We propose an algorithm, *Successive Refine Federated Clustering Algorithm* (`SR-FCA`), that treats each client as a singleton cluster as an initialization, and then successively refine the cluster estimation via exploiting similarity with other clients. In any intermediate step, `SR-FCA` uses an *error-tolerant* federated learning algorithm within each cluster to exploit simultaneous training and to correct clustering errors. Unlike some prominent prior works `SR-FCA` does not require any *good* initialization (or warm start), both in theory and practice. We show that with proper choice of learning rate, `SR-FCA` incurs arbitrarily small clustering error. Additionally, `SR-FCA` does not require the knowledge of the number of clusters apriori like some prior works. We validate the performance of `SR-FCA` on real-world FL datasets including FEMNIST and Shakespeare in non-convex problems and show the benefits of `SR-FCA` over several baselines.

## 1 Introduction

Federated Learning (FL), introduced in McMahan et al. (2016); Konečný et al. (2016); McMahan & Ramage (2017) is a large scale distributed learning paradigm aimed to exploit the machine intelligence in users' local devices. Owing to its highly decentralized nature, several statistical and computational challenges arise in FL, and in this paper, we aim to address one such challenge: heterogeneity.

The issue of heterogeneity is crucial for FL, since the data resides in users' own devices, and naturally no two devices have identical data distribution. There has been a rich body of literature in FL to address this problem of non-iid data. We direct the readers to two survey papers (and the references therein), Li et al. (2020a); Kairouz et al. (2019) for a comprehensive list of papers on heterogeneity in FL. A line of research assumes

the *degree of dissimilarity* across users is small, and hence focuses on learning a single global model Zhao et al. (2018); Li et al. (2020b; 2019); Sattler et al. (2019); Mohri et al. (2019); Karimireddy et al. (2020). Along with this, a line of research in FL focuses on obtaining models personalized to individual users. For example Li et al. (2020b; 2021) use a regularization to obtain individual models for users and the regularization ensures that the local models stay close to the global model. Another set of work poses the heterogeneous FL as a meta learning problem Chen et al. (2018); Jiang et al. (2019); Fallah et al. (2020b;a). Here, the objective is to first obtain a single global model, and then each device run some local iterations (fine tune) the global model to obtain their local models. Furthermore Collins et al. (2021) exploits shared representation across users by running an alternating minimization algorithm and personalization. Note that all these personalization algorithms, including meta learning, work only when the local models of the users' are close to one another (see bounded heterogeneity terms $\gamma_H$ and $\gamma_G$ terms in Assumption 5 of Fallah et al. (2020b)).

On the other spectrum, when the local models of the users may not be close to one another, Sattler et al. (2021); Mansour et al. (2020); Ghosh et al. (2022) propose a framework of *Clustered Federated Learning*. Here users with dissimilar data are put into different clusters, and the objective is to obtain individual models for each cluster; i.e., a joint training is performed within each cluster. Among these, Sattler et al. (2021) uses a top-down approach using cosine similarity metric between gradient norm as optimization objective. However, it uses a centralized clustering scheme, where the center has a significant amount of compute load, which is not desirable for FL. Also, the theoretical guarantees of Sattler et al. (2021) are limited. Further, in Duan et al. (2021), a data-driven similarity metric is used extending the cosine similarity and the framework of Sattler et al. (2021). Moreover, in Mansour et al. (2020), the authors propose algorithms for both clustering and personalization. However, they provide guarantees only on generalization, not iterate convergence. In Smith et al. (2017) the job of multi-task learning is framed as clustering where a regularizer in the optimization problem defines clustering objective.

Very recently, in Ghosh et al. (2022), an iterative method in the clustered federated learning framework called Iterative Federated Clustering Algorithm, or `IFCA`, was proposed and a *local convergence* guarantee was obtained. The problem setup for `IFCA` is somewhat restrictive—it requires the model (or data distribution) of all the users in the same cluster to be (exactly) identical. In order to converge, `IFCA` necessarily requires *suitable* initialization in clustering, which can be impractical. Furthermore, in Ghosh et al. (2022), all the users are partitioned into a fixed and known number of clusters, and it is discussed in the same paper that the knowledge about the number of clusters is quite non-trivial to obtain (see Section 6.3 in Ghosh et al. (2022)). There are follow up works, such as Ruan & Joe-Wong (2021), Xie et al. (2020), that extend `IFCA` in certain directions, but the crucial shortcomings, namely the requirements on *good* initialization and *identical* local models still remain unaddressed to the best of our knowledge.

In this paper, we address the above-mentioned shortcomings. We introduce a new clustering algorithm, Successive Refinement Federated Clustering Algorithm or `SR-FCA`, which leverages pairwise distance based clustering and refines the estimates over multiple rounds. We show that `SR-FCA` does not require any specific initialization. Moreover, we can allow the same users in a cluster to have non-identical models (or data distributions); in Section 2 we define a clustering structure (see Definition 2.1) that allows the models of the users in the same cluster to be different (we denote this discrepancy by parameter $\epsilon_1 (\geq 0)$. Furthermore, `SR-FCA` works with a different set of hyper-parameters which does not include the number of clusters and `SR-FCA` iteratively estimates this hyper-parameter.

**Clustering Framework and Distance Metric:** Classically, clustering is defined in terms of distribution from which the users sample data. However, in a federated framework, it is common to define a heterogeneous framework such as clustering in terms of other discrepancy metric; for example in Mansour et al. (2020), a metric that depends on the local loss is used.

In this paper, we use a distance metric across users' local model as a discrepancy measure and define a clustering setup based on this. Our distance metric may in general include non-trivial metric like Wasserstein distance, $\ell_q$ norm (with $q \geq 1$) that captures desired practical properties like permutation invariance and sparsity for (deep) neural-net training. For our theoretical results, we focus on strongly convex and smooth loss for which $\ell_2$ norm of iterates turns out to be the natural choice. However, for non-convex neural networks on which we run most of our experiments, we use a *cross-cluster loss* metric. For two clients $i, j$, we define their cross-cluster loss metric as the average of the loss of one client on the other's model, i.e., client $i$'s loss on the model of $j$ and the other way round. If this metric is low, we can use the model of client $i$ for client $j$ and vice-versa, implying

that the clients are similar. We explain this in detail in Section 5. With the above discrepancy metric, we put the users in same cluster if their local models are close – otherwise they are in different clusters.

## 1.1 Our Contributions

**Algorithms and Technical Contribution.** We introduce a novel clustering framework based on local user models and propose an iterative clustering algorithm, `SR-FCA`. Note that, since the clustering is defined based on the optimal models of the users, we have no way to know the clustering at the beginning of the process. To mitigate this, we start with initially assigning a different cluster for each user, and run few local iterations of SGD/GD in parallel. We then form a clustering based on the pairwise distance between iterates. This clustering is refined (including merges/splits if necessary) over multiple rounds of our algorithm. We run federated training on each of the clusters to further improve the models. This step exploits collaboration across users in the same cluster. However, clustering based on iterates might lead to many mis-clustered clients, as the iterates might be far from optimal models. The mis-clustered clients might lead to more errors from ordinary federated training within clusters (error propagation) due to high heterogeneity between the original clusters.

To counter this we run a *robust* federated training (based on trimmed mean) within each clusters in the intermediate rounds, instead of straightforward federated learning. In particular, we use the first order gradient based robust FL algorithm of Yin et al. (2018) to handle the clustering error. Within a cluster, we treat the wrongly clustered users as outliers. However, instead of throwing the outliers away like Yin et al. (2018), we reassign them to their closest cluster.

When the loss is strongly convex and smooth, and $\mathsf{dist}(.,.)$ is $\ell_2$ norm, we show that, the mis-clustering error in the first stage of `SR-FCA` is given by $\mathcal{O}(md\exp(-n/\sqrt{d}))$ ( Lemma 4.6), where $m$, $n$ and $d$ denote the number of users, the amount of data in each user and the dimensionality of the problem respectively. Moreover, successive stages of `SR-FCA` further reduce the mis-clustering error by a factor of $\mathcal{O}(1/m)$ (Theorem 4.8), and hence yields arbitrarily small error. In practice we require very few refinement steps (we refine at most twice in experiments, see Section 5). Comparing our results with `IFCA` Ghosh et al. (2022), we notice that the requirement on the separation of clusters is quite mild for `SR-FCA`. We only need the separation to be[1] $\tilde{\Omega}(\frac{1}{n})$. On the other hand, in certain regimes, `IFCA` requires a separation of $\tilde{\Omega}(\frac{1}{n^{1/5}})$, which is a much stronger requirement.

To summarize, a key assumption in any clustering problem (which is non-convex) is *suitable* initialization. However, `SR-FCA` removes this requirement completely by the above technique, and allows the clients to start arbitrarily. For our results, we crucially leverage (a) sharp generalization guarantees for strongly convex losses with sub-exponential gradients and (b) robustness property of the trimmed mean estimator (of Yin et al. (2018)).

As a by-product of `SR-FCA`, we also obtain an appropriate loss minimizer for each cluster ( Theorem 4.14). We notice that the statistical error we obtain here is $\tilde{\mathcal{O}}(1/\sqrt{n})$. This statistical rate primarily comes from the usage of the robust estimator of Yin et al. (2018).

**Experiments.** We implement `SR-FCA` on wide variety of simulated heterogeneous datasets (rotated or inverted MNIST, CIFAR10) and real federated datasets (FEMNIST and Shakespeare Caldas et al. (2018)). With *cross-cluster* loss distance metric , we compare the test performance of `SR-FCA` with five baselines—(a) global (one model for all users) and (b) local (one model per user), (c)`IFCA`, (d) CFL Sattler et al. (2021) and (e) Local-KMeans (local models clustered by KMeans on model weights). On simulated datasets, `SR-FCA` obtains test accuracy no worse than the best baseline and is able to recover the correct clustering. For CIFAR10, in particular, `SR-FCA` has 5% better test accuracy than `IFCA`. On real datasets, `SR-FCA` outperforms all baselines.

## 2 Federated Clustering and Our Setup

In this section, we formally define the clustering problem. Let, $[n] \equiv \{1,2,...,n\}$. We have $m$ users (or machines) that are partitioned into disjoint clusters, denoted by the clustering map $\mathcal{C}^\star : [m] \to [K]$, where $K$ is the (unknown) number of clusters. Let $\mathcal{C}' : [m] \to \mathrm{rg}(\mathcal{C}')$ denote any arbitrary clustering map $\mathcal{C}'$, where $\mathrm{rg}(\mathcal{C}')$ is the range of the clustering. Each user $i \in [m]$ contains $n_i \geq n$ data points $\{z_{i,j}\}_{j=1}^{n_i}$ sampled from a distribution $\mathcal{D}_i$. For any clustering map $\mathcal{C}'$, let $\mathrm{rg}(\mathcal{C}')$ denote the range of the map. We define $f(\cdot;z) : \mathcal{W} \to \mathbb{R}$ as the loss

---

[1]Here, $\tilde{\mathcal{O}}$ and $\tilde{\Omega}$ hide logarithmic dependence.

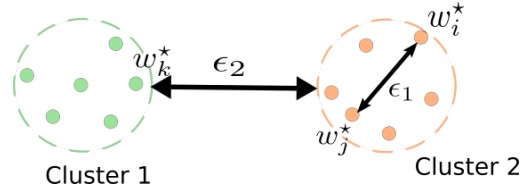

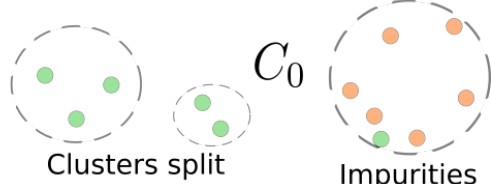

Figure 1: The dots represent the population risk minimizers for two clusters in $\mathsf{dist}(.,.)$ space according to $\mathcal{C}^\star$.

Figure 2: The dots represent the ERM in $\mathsf{dist}(.,.)$ space and the corresponding clustering $\mathcal{C}_0$ obtained after ONE_SHOT

function for the sample $z$, where $\mathcal{W} \subseteq \mathbb{R}^d$. Here, $\mathcal{W}$ is a closed and convex set with diameter $D$. We use $\mathcal{C}$ to denote different clustering maps on the set of clients $[m]$ encountered by our algorithm. We use $\mathrm{rg}(\mathcal{C})$ to denote the range of the clustering map, i.e., the cluster indices. We define the population loss, $F_i : \mathcal{W} \to \mathbb{R}^d$, and its minimizer, $w_i^\star$, for each user $i \in [m]$: $F_i(w) = \mathbb{E}_{z \sim \mathcal{D}_i}[f(w,z)]$, $\quad w_i^\star = \min_{w \in \mathcal{W}} F_i(w)$. The original clustering $\mathcal{C}^\star$ is based on the population minimizers of users, $w_i^\star$. This is defined as:

**Definition 2.1** (Clustering Structure). For a distance metric $\mathsf{dist}(.,.)$, the local models satisfy

$$\max_{i,j \in [m]: \mathcal{C}^\star(i) = \mathcal{C}^\star(j)} \mathsf{dist}(w_i^\star, w_j^\star) \leq \epsilon_1, \quad \min_{i,j \in [m]: \mathcal{C}^\star(i) \neq \mathcal{C}^\star(j)} \mathsf{dist}(w_i^\star, w_j^\star) \geq \epsilon_2. \tag{1}$$

where $\epsilon_1, \epsilon_2$, are non-negative constants with $\epsilon_2 > \epsilon_1$. This is illustrated in fig. 1.

The above allows the population minimizers inside clusters to be close, but not necessarily equal.

Let $G_c \equiv \{i : i \in [m], \mathcal{C}^\star(i) = c\}$ denote the set of users in cluster $c$ according to the original clustering $\mathcal{C}^\star$. We can then define the population loss and its minimizer, per cluster $c \in [K]$ as follows,

$$\mathcal{F}_c(w) = \frac{1}{|G_c|} \sum_{i \in G_c} F_i(w), \quad \omega_c^* = \operatorname*{argmin}_{w \in \mathcal{W}} \mathcal{F}_c(w) \tag{2}$$

We use $w_{i,T}$ for every client $i \in [m]$ to denote the local model on client $i$ obtained after $T$ local iterations of a Our goal is to find a population loss minimizer for each cluster $c \in [K]$, i.e., $\omega_c^*$. To obtain this, we need to find the correct clustering $\mathcal{C}^\star$ and recover the minimizer of each cluster's population loss. Note that we have access to neither $F_i$ nor $w_i^\star$, but only the sample mean variant of the loss, the empirical risk, $f_i(w) = \frac{1}{n_i} \sum_{j=1}^{n_i} f(w, z_{i,j})$ for each user $i \in [m]$. There are two major difficulties in this setting: (a) the number of clusters is not known beforehand. This prevents us from using most clustering algorithms like $k$-means, and (b) The clustering depends on $w_i^\star$ which we do not have access to. We can estimate $w_i^\star$ by minimizing $f_i$, however, when $n$, the minimum number of data points per user, is small, this estimate may be very far from $w_i^\star$.

The above difficulties can be overcome by utilizing federation. First, instead of estimating $w_i^\star$ for a single user, we can estimate $\omega_c^\star$, the population minimizer for each cluster, where users in the cluster collaborate to improve the estimate. Second, we can use these estimates of $\omega_c^\star$ to improve the clustering, according to Definition 3.1.

## 3 Algorithm : SR-FCA

In this section, we formally present our clustering algorithm, SR-FCA. We first run the subroutine ONE_SHOT to obtain an appropriate initial clustering, which can be further improved. SR-FCA then successively calls the REFINE() subroutine to improve this clustering. In each step of REFINE(), we first estimate the cluster models for each cluster. Then, based on these models we regroup all the users using RECLUSTER() and, if required, we merge the resulting clusters, using MERGE(). The full algorithm along with its subroutines is provided in Algorithms 1-4. ONE_SHOT and TrimmedMeanGD can be decomposed into server and client subroutines. For the clustering subroutines, RECLUSTER and MERGE, all steps are performed on the server, barring the computation of $\mathsf{dist}$ which might require communication between the server and the clients. We now explain the different subroutines in detail.

### Algorithm 1: SR-FCA

**Input:** Threshold $\lambda$, Size parameter $t$
**Output:** Clustering $\mathcal{C}_R$
$\mathcal{C}_0 \leftarrow \texttt{ONE\_SHOT}(\lambda, t)$
**for** $r = 1$ to $R$ **do**
    $\mathcal{C}_r \leftarrow \texttt{REFINE}(\mathcal{C}_{r-1}, \lambda)$
**end for**

$\underline{\texttt{ONE\_SHOT}(\lambda, t)}$
**Server:**
**for** all $i$ clients in parallel **do**
    Send $w_0$ to client $i$
    Receive $w_{i,T}$ from client $i$
**end for**
$G \leftarrow$ Graph with $m$ nodes (one per user) and no edges
**for** all pairs of clients $i,j \in [m], i \neq j$ **do**
    Add edge $(i,j)$ to the graph $G$ if $\mathsf{dist}(w_{i,T}, w_{j,T}) \leq \lambda$
**end for**
$\mathcal{C}_0 \leftarrow$ Connected components from graph $G$ with size $\geq t$
**Client($i$):**
Receive $w_0$ from Server.
$w_{i,T} \leftarrow$ Perform $T$ training iterations initialized from $w_0$.
Send $w_{i,T}$ to Server.

$\underline{\texttt{REFINE}(\mathcal{C}_{r-1}, \lambda)}$
**for** all clusters $c \in \mathcal{C}_{r-1}$ **do**
    $\omega_{c,T} \leftarrow \texttt{TrimmedMeanGD}()$
**end for**
$\mathcal{C}'_r \leftarrow \texttt{RECLUSTER}(\mathcal{C}_{r-1})$
$\mathcal{C}_r \leftarrow \texttt{MERGE}(\mathcal{C}'_r, \lambda, t)$

### Algorithm 2: RECLUSTER()

**Input:** Cluster models $\{\omega_{c,T}\}_{c \in \mathrm{rg}(\mathcal{C}_r)}$, User models $\{w_i\}_{i=1}^m$, Clustering $\mathcal{C}_r$
**Output:** Improved Clustering $\mathcal{C}'_r$
**for** all users $i \in [m]$ **do**
    $\mathcal{C}'_r(i) \leftarrow \mathrm{argmin}_{c \in \mathrm{rg}(\mathcal{C}_r)} \mathsf{dist}(w_i, \omega_{c,T})$
**end for**
**return** Clustering $\mathcal{C}'_r$.

### Algorithm 3: TrimmedMeanGD()

**Input:** $0 \leq \beta < \frac{1}{2}$, Clustering $\mathcal{C}_r$
**Output:** Cluster models $\{\omega_{c,T}\}_{c \in \mathrm{rg}(\mathcal{C}_r)}$
**Server:**
**for** all clusters $c \in \mathrm{rg}(\mathcal{C}_r)$ in parallel **do**
    $\mathcal{S}_c = \{i \in [m] : \mathcal{C}_r(i) = c\}$.
    $\omega_{c,0} \leftarrow w_0$
    **for** $t = 0$ to $T-1$ **do**
        Send $\omega_{c,t}$ to all clients $i \in \mathcal{S}_c$.
        Receive $\nabla f_i(\omega_{c,t})$ from all clients $i \in \mathcal{S}_c$.
        $g(\omega_{c,t}) \leftarrow \mathrm{TrMean}_\beta(\{\nabla f_i(w_{c,t}), i \in \mathcal{S}_c\})$
        $\omega_{c,t+1} \leftarrow proj_\mathcal{W}\{\omega_{c,t} - \eta g(\omega_{c,t})\}$
    **end for**
**end for**
**Return** $\{\omega_{c,T}\}_{c \in \mathrm{rg}(\mathcal{C}_r)}$
**Client($i$):**
Receive $\omega_{c,T}$ from Server.
Send $\nabla f_i(\omega_{c,T})$ to Server.

### Algorithm 4: MERGE()

**Input:** Cluster models $\{\omega_{c,T}\}_{c \in \mathrm{rg}(\mathcal{C}_r)}$, Clustering $\mathcal{C}'_r$, Threshold $\lambda$, Size parameter $t$
**Output:** Merged Clustering $\mathcal{C}_{r+1}$, Cluster models $\{\omega_{c,T}\}_{c \in \mathrm{rg}(\mathcal{C}_{r+1})}$
$G \leftarrow$ Graph with nodes $\mathrm{rg}(\mathcal{C}'_r)$ and no edges
**for** all pairs of clusters $c,c' \in \mathrm{rg}(\mathcal{C}'_r), c \neq c'$ **do**
    Add edge $(c,c')$ to the graph $G$ if $\mathsf{dist}(w_c, w_{c'}) \leq \lambda$
**end for**
$\mathcal{C}_{temp} \leftarrow$ Connected components from graph $G$ of size $\geq t$
For each cluster in $\mathcal{C}_{temp}$, merge the nodes of its component clusters to get $\mathcal{C}_{r+1}$
**for** $c \in \mathrm{rg}(\mathcal{C}_{temp})$ **do**
    $G_c \leftarrow \{c' \in \mathrm{rg}(\mathcal{C}'_r)$ which merged into $c\}$
    $\omega_{c,T} \leftarrow \frac{1}{|G_c|} \sum_{c' \in G_c} \omega_{c',T}$
**end for**
**return** $\mathcal{C}_{r+1}, \{\omega_{c,T}\}_{c \in \mathrm{rg}(\mathcal{C}_{r+1})}$.

## 3.1 ONE_SHOT()

For our initial clustering, we create edges between users based on the distance between their locally trained models if $\mathsf{dist}(w_i, w_j) \leq \lambda$, for a threshold $\lambda$. We obtain clusters from this graph by simply finding the connected components, which can be done in time linear in number of edges. We only keep the clusters which have at least $t$ users.

We use $w_{i,T}$ to denote the model obtained after $T$ local iterations of any optimizer on client $i \in [m]$. If our locally trained models, $w_{i,T}$, were close to their population minimizers, $w_i^\star$, for all users $i \in [m]$, then choosing a threshold $\lambda \in (\epsilon_1, \epsilon_2)$, we obtain edges between only clients which were in the same cluster in $\mathcal{C}^\star$. However, if $n$, the number of local datapoints is small, then our estimates of local models $w_{i,T}$ might be far from their corresponding $w_i^\star$ and we will not be able to recover $\mathcal{C}^\star$.

However, $\mathcal{C}_0$ is still a good clustering if it satisfies these requirements: (a) if every cluster in the range of the clustering map $\mathrm{rg}(\mathcal{C}^\star) = [K]$ has a good proxy (in the sense of definition 3.1) in $\mathrm{rg}(\mathcal{C}_0)$, and (b) each cluster in $\mathrm{rg}(\mathcal{C}_0)$ has at most a small fraction ($< \frac{1}{2}$) of mis-clustered users in it. E.g., fig. 2 provides an example of one such good clustering when $\mathcal{C}^\star$ is defined according to fig. 1. We can see that even though $\mathcal{C}_0 \neq \mathcal{C}^\star$, the two green clusters and the single orange cluster in $\mathcal{C}_0$ are mostly "pure" and are proxies of Cluster 1 and Cluster 2 in fig. 1.

To formally define the notion of "purity" and "proxy", we introduce the notion of cluster label for any arbitrary clustering $\mathcal{C}'$, which relates it to the original clustering $\mathcal{C}^\star$.

**Definition 3.1** (Cluster label). We define $c \in [K]$, as the cluster label of cluster $c' \in \mathrm{rg}(\mathcal{C}')$ if the majority ($> 1/2$ fraction) of users in $c'$ are originally from $c$.

This definition allows us to map each cluster $c' \in \mathrm{rg}(\mathcal{C}')$ to a cluster $c$ in $\mathcal{C}^\star$ and thus define the notion of "proxy". In fig. 2, the cluster label of green clusters is Cluster 1 and that of orange cluster is Cluster 2. Further, using the cluster label $c$, we can define the impurities in cluster $c'$ as the users that did not come from $c'$. In fig. 2, the green node in orange cluster is an impurity. Based on these definitions, we can see that if clusters in $\mathcal{C}_0$ are mostly pure and can represent all clusters in $\mathcal{C}^\star$, then $\mathcal{C}_0$ is a good clustering.

## 3.2 `REFINE()`

We iteratively refine the clustering obtained by `ONE_SHOT()` using `REFINE()`. We describe the subroutines of a single `REFINE` step below.

**Subroutine `TrimmedMeanGD()`.** The main issue with `ONE_SHOT()`, namely, small $n$, can be mitigated if we use federation. Since $\mathcal{C}_0$ has atleast $t$ users per cluster, training a single model for each cluster will utilize $\geq tn$ datapoints, making the estimation more accurate. However, from fig. 2, we can see that the clusters contain impurities, i.e., users from a different cluster. To handle them, we use a robust training algorithm, TrimmedMeanYin et al. (2018).

This subroutine is similar to FedAvg McMahan et al. (2016), but instead of taking the average of local models, we take the coordinate-wise trimmed mean, referred to as $\mathrm{TrMean}_\beta$ where $\beta \in (0, 1/2)$ defines the trimming level.

**Definition 3.2** ($\mathrm{TrMean}_\beta$). For $\beta \in [0, \frac{1}{2})$, and a set of vectors $x^j \in \mathbb{R}^d, j \in [J]$, their coordinate-wise trimmed mean $g = \mathrm{TrMean}_\beta(\{x^1, x^2, ..., x^J\})$ is a vector $g \in \mathbb{R}^d$, with each coordinate $g_k = \frac{1}{(1-2\beta)J} \sum_{x \in U_k} x$, for each $k \in [d]$, where $U_k$ is a subset of $\{x_k^1, x_k^2, ..., x_k^J\}$ obtained by removing the smallest and largest $\beta$ fraction of its elements.

The full algorithm for `TrimmedMeanGD` is provided in algorithm 3. Note that $\mathrm{TrMean}_\beta$ has been used to handle Byzantine users, achieving optimal statistical rates Yin et al. (2018), when $< \beta$ fraction of the users are Byzantine. For our problem setting, there are no Byzantine users as such and we use $\mathrm{TrMean}_\beta$ to handle users from different clusters as impurities.

Note the two requirements for good clustering $\mathcal{C}_0$ from `ONE_SHOT`: (a) if every cluster in $\mathcal{C}^\star$ has a proxy in $\mathcal{C}_0$, then the `TrimmedMeanGD` obtains at least one cluster model for every cluster in $\mathcal{C}^\star$, (b) if every cluster in $\mathcal{C}_0$ has a small fraction ($\beta < \frac{1}{2}$) of impurities, then we can apply $\mathrm{TrMean}_\beta$ operation can recover the correct cluster model for every cluster.

We end up with a trained model for each cluster as an output of this subroutine. Since these models are better estimates of their population risk minimizers than before, we can use them to improve $\mathcal{C}_0$.

**Subroutine `RECLUSTER()`.** The full algorithm for this subroutine is provided in algorithm 2. This subroutine reduces the impurity level of each cluster in $\mathcal{C}_0$ by assigning each client $i$ to its nearest cluster $c$ in terms of $\mathrm{dist}(\omega_{c,T}, w_{i,T})$. Here, $\omega_{c,T}$ refers to the model on cluster $c$ obtained after `TrimmedMeanGD` with $T$ iterations. Since $\omega_{c,T}$ are better estimates, we hope that the each impure user will go to a cluster with its actual cluster label. For instance, in fig. 2, the impure green node should go to one of the green clusters. If some clusters in $\mathrm{rg}(\mathcal{C}^\star)$ do not have a good proxy in $\mathrm{rg}(\mathcal{C}_0)$, then the users of this cluster will always remain as impurities.

**Subroutine `MERGE()`.** We provide the full algorithm for this subroutine in algorithm 4. Even after removing all impurities from each cluster, we can still end up with clusters in $\mathcal{C}^\star$ being split, for instance the green clusters in fig. 2. In $\mathcal{C}^\star$, these form the same cluster, thus they should be merged. As these were originally from the same cluster in $\mathcal{C}^\star$, their learned models should also be very close. Similar to `ONE_SHOT`, we create a graph $G$ but instead with nodes being the clusters in $\mathcal{C}'_r$. Then, we add edges between clusters based on a threshold $\lambda$ and find all the clusters in the resultant graph $G$ by finding all connected components. Then, each of these clusters in $G$ correspond to a set of clusters in $\mathcal{C}'_r$, so we merge them into a single cluster to obtain the final clustering $\mathcal{C}_{r+1}$.

### 3.3 Discussion

`SR-FCA` uses a bottom-up approach to construct and refine clusters. The initialization in `ONE_SHOT` is obtained by distance-based thresholding on local models. These local models are improper estimates of their population minimizers due to small $n$, causing $\mathcal{C}_0 \neq \mathcal{C}^\star$. However, if $\mathcal{C}_0$ is not very bad, i.e., each cluster has $< \frac{1}{2}$ impurity fraction and all clusters in $\mathcal{C}^\star$ are represented, we can refine it.

`REFINE()` is an alternating procedure, where we first estimate cluster centers from impure clusters. Then, we `RECLUSTER()` to remove the impurities in each cluster and then `MERGE()` the clusters which should be merged according to $\mathcal{C}^\star$. Note that as these steps use more accurate cluster estimates, they should have smaller error. This iterative procedure should recover one cluster for each cluster in $\mathcal{C}^\star$, thus obtaining the number of clusters and every cluster should be pure so that $\mathcal{C}^\star$ is exactly recovered.

### 3.4 Computation and Communication Complexity

We now analyze the computation and communication complexity of `SR-FCA`. Note that the complexity of the `REFINE` step is the same as that of `IFCA` in terms of both computation time and communication since in each case, we need to find the loss of every cluster model on every client's data (all pairs). This requires $\mathcal{O}(m^2)$ forward passes. Note that performing all pairwise comparisons is unavoidable if an initial clustering is not known, as we need to know which clients can be clustered together. (eg. see KMeans Lloyd (1982) v/s DBSCAN Ester et al. (1996) or Ward's algorithm).

The `REFINE` step is comparable to `IFCA` for which we provide a detailed analysis. Assume that we run `IFCA` on $K$ clusters for $T$ rounds with $E$ local steps with $\alpha$ fraction of $m$ clients participating in each round. Then, we require $\mathcal{O}(\alpha m T E)$ backward passes and $\mathcal{O}(\alpha m T)$ communication for local training and aggregation. For cluster identity estimation, in each round, we need to compute the loss of each cluster model on every participating client. To send cluster models to each client, we need $\mathcal{O}(\alpha m K T)$ communication and to compute losses, we need $\mathcal{O}(\alpha m K T)$ forward passes.

If we run $R$ `REFINE` steps with $\leq C$ clusters per step where each `TrimmedMeanGD` procedure runs for $T$ rounds with $E$ local steps and $\alpha$ fraction of clients participating in each round. We need $\mathcal{O}(\alpha m E T R)$ backward passes and $\mathcal{O}(\alpha m T R)$ communication for aggregation and local training. For clustering in `REFINE` and `MERGE` steps, we need to compute the loss of every model on every client, which requires $\mathcal{O}(mKR)$ communication and $\mathcal{O}(mKR)$ forward passes. We summarize these results in Table 1.

| Algorithm | Communication | Training Steps | Forward Passes |
|---|---|---|---|
| IFCA | $\mathcal{O}(\alpha m T K)$ | $\mathcal{O}(\alpha m T E)$ | $\mathcal{O}(\alpha m T K)$ |
| SR-FCA | $\mathcal{O}(\alpha m T R + m K R)$ | $\mathcal{O}(\alpha m T E R)$ | $\mathcal{O}(m K R)$ |

Table 1: Communication, training runtime and forward passes for loss computation in clustering for `SR-FCA` and `IFCA`. $E$ is the number of local steps, $T$ is the number of rounds, $R$ is the number of `REFINE` steps, $K$ is the number of clusters and $\alpha$ is the fraction of clients participating per round.

Comparing the two algorithms, where the number of `REFINE` steps $R$ and the number of clusters $C$ are assumed to be constants, for a constant $\alpha$, both algorithms require the same communication and backward passes, but `IFCA` needs more forward passes for clustering. If $\alpha = \Theta(\frac{1}{m})$, which corresponds to selecting a constant number of clients per round, then if $m = \Omega(T)$, `IFCA` is better than `SR-FCA` in terms of communication and forward passes, and vice-versa if $T = \Omega(m)$.

As for the other baselines, Local-KMeans is the most efficient in terms of computation and communication complexity as it performs only a single round of communication, at the end of training. CFL, on the other hand, adopts a top-down approach to split clusters into two parts based on the cosine similarity of gradients. Each split calls a bipartitioning subroutine with runtime $\mathcal{O}(m^3)$ making it the slowest baseline.

## 4 Theoretical Guarantees

In this section, we obtain the convergence guarantees of `SR-FCA`. For theoretical tractability, we impose additional conditions on `SR-FCA`. First, the dist(.,.) is the Euclidean ($\ell_2$). However, in experiments (see next section), we remove this restriction and work with other dist(.,.) functions. Here, we show an example where $\ell_2$ norm comes naturally as the dist(.,.) function.

**Proposition 4.1.** *Suppose that there are $m$ users, each with a local model $w_i^\star \in \mathbb{R}^d$ and its datapoint $(x, y_i) \in \mathbb{R}^d \times \mathbb{R}$ is generated according to $y_i = \langle w_i^\star, x \rangle + \epsilon_i$. If $x \sim \mathcal{N}(0, I_d)$ and $\epsilon_i \overset{i.i.d}{\sim} \mathcal{N}(0, \sigma^2)$, then $KL(p(x, y_i) || p(x, y_j)) = \mathbb{E}_x[KL(p(y_i|x) || p(y_j|x))] = \frac{d}{2\sigma^2} \|w_i - w_j\|^2$.*

Hence, we see that minimizing a natural measure (KL divergence) between the distributions for different users is equivalent to minimizing the $\ell_2$ distance of the underlying local models. This example only serves as a motivation, and our theoretical results hold for a strictly larger class of functions, as defined by our assumptions.

*Remark* 4.2 ($\lambda$ range). For the guarantees of this section to hold, we require $\lambda \in (\epsilon_1, \epsilon_2)$ and $t \le c_{\min}$, where $c_{\min}$ is the minimum size of the cluster. We emphasize that, in practice (as shown in the experiments), we treat $\lambda$ and $t$ as hyper-parameters and obtain them by tuning. Hence, we do not require the knowledge of $\epsilon_1, \epsilon_2$ and $c_{\min}$.

The following assumptions specify the exact class of losses for which our analysis holds. Definitions provided in appendix E.

**Assumption 4.3** (Strong convexity). The loss per sample $f(w,.)$ is $\mu$-strongly convex with respect to $w$.

**Assumption 4.4** (Smoothness). The loss per sample $f(w,.)$ is also $L$-smooth with respect to $w$.

**Assumption 4.5** (Lipschitz). The loss per sample $f(w,.)$ is $L_k$-Lipschitz for every coordinate $k \in [d]$. Define $\hat{L} = \sqrt{\sum_{k=1}^d L_k^2}$.

We want to emphasize that the above assumptions are standard and have appeared in the previous literature. For example, the strong convexity and smoothness conditions are often required to obtain theoretical guarantees for clustering models (see Ghosh et al. (2022); Lu & Zhou (2016), including the classical $k$-means which assume a quadratic objective. The coordinate-wise Lipschitz assumption is also not new and (equivalent assumptions) featured in previous works Yin et al. (2018; 2019), with it being necessary to establish convergence of the trimmed mean procedure. Throughout this section, we require Assumption 4.3, 4.4 and 4.5 to hold.

**Misclustering Error** Since the goal of `SR-FCA` is to recover both the clustering and cluster models, we first quantify the probability of not recovering the original clustering, i.e., $\mathcal{C}_r \neq \mathcal{C}^\star$. Here and subsequently, two clusters being not equal means they are not equal after relabeling (see definition 3.1). We are now ready to show the guarantees of several subroutines of `SR-FCA`. First, we show the probability of misclustering after the `ONE_SHOT` step.

**Lemma 4.6** (Error after `ONE_SHOT`). *After running `ONE_SHOT` with $\eta \le \frac{1}{L}$ for $T$ iterations, for the threshold $\lambda \in (\epsilon_1, \epsilon_2)$ and some constant $b_2 > 0$, the probability of error is $\Pr[\mathcal{C}_0 \neq \mathcal{C}^\star] \le p \equiv md \ \exp(-n\frac{b_2\Delta}{\hat{L}\sqrt{d}})$, provided $\frac{n^{2/3}\Delta^{4/3}}{D^{2/3}\hat{L}^{2/3}} \gtrsim d$, where $\Delta = \frac{\mu}{2}(\frac{\min\{\epsilon_2-\lambda, \lambda-\epsilon_1\}}{2} - (1-\frac{\mu}{L})^{T/2}D)$ and $n = \min_{i \in [m]} n_i$.*

We would like to emphasize that the probability of error is exponential in $n$, yielding a *reasonable* good clustering after the `ONE_SHOT` step. Note that the best probability of error is obtained when $\lambda = \frac{\epsilon_1+\epsilon_2}{2}$.

*Remark* 4.7 (Separation). In order to obtain $p < 1$, we require $\Delta = \Omega(\frac{\log m}{n})$. Since $\Delta \le \frac{\mu}{2}\frac{\epsilon_2-\epsilon_1}{4}$, we require $(\epsilon_2 - \epsilon_1) \ge \mathcal{O}(\frac{\log m}{n}) = \tilde{\mathcal{O}}(\frac{1}{n})$. Note that we require a condition only on the separation $\epsilon_2 - \epsilon_1$, instead of just $\epsilon_2$ or $\epsilon_1$ individually.

Although we obtain an exponentially decreasing probability of error, we would like to improve the dependence on $m$, the number of users. `REFINE()` step does this job.

**Theorem 4.8** (One step `REFINE()`). *Let $\beta t = \Theta(c_{\min})$, and `REFINE()` is run with `TrimmedMeanGD($\beta$)`. Provided $\min\{\frac{n^{2/3}\Delta'^{4/3}}{D^{2/3}}, \frac{n^2\Delta'^2}{\hat{L}^2\log(c_{\min})}\} \gtrsim d$, with $0 < \beta < \frac{1}{2}$, where $\Delta' = \Delta - \frac{\mu B}{2} > 0$ and $B = \sqrt{2\hat{L}\epsilon_1/\mu}$. Then, for any*

*constant* $\gamma_1 \in (1,2)$ *and* $\gamma_2 \in (1, 2 - \frac{\mu B}{2\Delta})$, *such that after running 1 step of* `REFINE()` *with* $\eta \leq \frac{1}{L}$, *we have*

$$\Pr[\mathcal{C}_1 \neq \mathcal{C}^\star] \leq \frac{m}{c_{\min}} \exp(-a_1 c_{\min}) + \frac{m}{t} \exp(-a_2 m) + (1-\beta)m(\frac{p}{m})^{\gamma_1} + m(\frac{p}{m})^{\gamma_2} + 8d\frac{m}{t}\exp(-a_3 n \frac{\Delta'}{2\hat{L}})$$

*where* $c_{\min}$ *is the minimum size of the cluster. Further for some small constants* $\rho_1 > 0, \rho_2 \in (0,1)$, *we can select* $\beta, \gamma_1$ *and* $\gamma_2$ *such that for large* $m, n$ *and* $\Delta'$, *with* $B << \frac{2\Delta'}{\mu}$, *we have* $\Pr[\mathcal{C}_1 \neq \mathcal{C}^\star] \leq \frac{\rho_1}{m^{1-\rho_2}} p$.

*Remark* 4.9 (Misclustering error improvement). Note that $\rho_2$ can be made arbitrarily close to 0 by a proper choice of $\gamma_1$ and $\gamma_2$. So, one step of `REFINE()` brings down the misclustering error by (almost) a factor of $1/m$, where $m$ is the number of users.

*Remark* 4.10 (Condition on $B$). Note that we require $B << \frac{2\Delta'}{\mu}$ for the above to hold. From the definition of $B$, when the intra-cluster separation $\epsilon_1$ is small, $B$ is small. So, for a setup like `IFCA`, where $\epsilon_1 = 0$, this condition is automatically satisfied.

**Proof Sketch for Theorem 4.8:** With the notation, $\mathcal{C}_1 \neq \mathcal{C}^\star$, we can identify the main sources of error for this event as: (1) *There exists* $c \in \mathrm{rg}(\mathcal{C}^\star)$ *such that no cluster in* $\mathcal{C}_0$ *has cluster label* $c$. (2) *Each cluster* $c \in \mathrm{rg}(C_0)$ *should have* $< \alpha$ *fraction of impurities for some* $\frac{1}{2} > \beta > \alpha$. (3)`MERGE()` *error:* Two clusters with same label are not merged or two clusters with different labels are merged. (4)`RECLUSTER()` *error:* A client does not go to its correct cluster after `MERGE()` and `REFINE()`.

For the first two sources of error, we utilize the clustering structure obtained by `ONE_SHOT`, while for the last two sources of error we leverage strong concentration of coordinate-wise Lipschitz losses to bound the performance of improved cluster models obtained by `TrimmedMeanGD`. In appendix C, we upper bound the probabilities of error for each of the above mentioned cases and apply a union bound on these errors to obtain the final result.

Using single step improvement of `REFINE`, we obtain the improvement after $R$ steps of `REFINE`.

**Theorem 4.11** (Multi-step `REFINE()`). *If we run $R$ steps of `REFINE()`, resampling $n_i$ points from $\mathcal{D}_i$ and recompute $w_i$ as in `ONE_SHOT` for every step of `REFINE()`, then the probability of error for `SR-FCA` with $R$ steps of `REFINE()` is* $\Pr[\mathcal{C}_R \neq \mathcal{C}^\star] \leq \left(\frac{\rho_2}{m^{(1-\rho_1)}} p\right)^R$.

*Remark* 4.12 (Re-sampling). Although the theoretical convergence of Multi-step `REFINE()` requires resampling of data points in each iteration of `REFINE()`, we experimentally validate (see section 5, that this is not required at all.

*Remark* 4.13. Since each step of `REFINE()` reduces the probability of misclusteing by (almost) a factor of $1/m$, very few steps of `REFINE()` is often sufficient. In our experiments ( section 5), we need $1-2$ `REFINE()` steps.

**Convergence of cluster models:** `SR-FCA` also obtain an appropriate loss minimizer for each cluster.

**Theorem 4.14** (Cluster models). *Under the conditions described in theorem 4.8, after running `SR-FCA` for* $(R+1)$ *steps of `REFINE()`, we have* $\mathcal{C}^{R+1} = \mathcal{C}^\star$ *and*

$$\|\omega_{c,T} - \omega_c^\star\| \leq (1 - \kappa^{-1})^{T/2} D + \Lambda + 2B, \text{where, } \Lambda = \mathcal{O}\left(\frac{\hat{L}d}{1-2\beta}\left(\frac{\beta}{\sqrt{n}} + \frac{1}{\sqrt{nc_{\min}}}\right)\sqrt{\log(nm\hat{L}D)}\right)$$

$\forall c \in \mathrm{rg}(\mathcal{C}^\star)$, *with probability* $1 - \left(\frac{\rho_2}{m^{(1-\rho_1)}} p\right)^R - \frac{m}{c_{\min}} \frac{4du''}{(1+nc_{\min}\hat{L}D)^d}$, *for some constant* $u'' > 0$.

*Remark* 4.15 (Convergence rate matches `IFCA`). Note that the models converge exponentially fast to the true cluster parameter $\omega_c^\star$, which matches the convergence speed of `IFCA`.

*Remark* 4.16 (Comparison with `IFCA` in statistical error). Note that for `IFCA`, $\epsilon_1 = 0$ and the statistical error rate of `IFCA` is $\tilde{\mathcal{O}}(1/n)$ (see Theorem 2 in Ghosh et al. (2022)). Looking at theorem 4.14, we see that under similar condition ($\epsilon_1 = 0$ and hence $B = 0$), `SR-FCA` obtains an error rate of $\tilde{\mathcal{O}}(1/\sqrt{n})$, which is weaker than `IFCA`. This can be thought of as the price of initialization. In fact for `IFCA`, a *good* initialization implies that only a very few users will be mis-clustered, which was crucially required to obtain the $\tilde{\mathcal{O}}(1/n)$ rate. But, for `SR-FCA`, we do not have such guarantees which results in a weaker statistical error.

## 5 Experiments

We compare the performance of `SR-FCA` against several baselines on simulated and real datasets.

Table 2: Test Accuracy and standard deviations across 5 random seeds on simulated datasets. The highest accuracy is **bold**. `SR-FCA` is competitve with `IFCA` for MNIST and beats it for CIFAR10.

| BASELINE | MNIST (INVERTED) | MNIST (ROTATED) | CIFAR (ROTATED) | CIFAR (LABEL) |
|---|---|---|---|---|
| SR-FCA | **92.03 ±0.30** | **91.66 ± 0.13** | **91.38 ± 0.27** | **93.06 ± 0.20** |
| LOCAL | 76.52 ±0.54 | 85.55 ± 0.19 | 75.87± 0.33 | 80.09 ± 0.53 |
| GLOBAL | 88.61 ± 0.77 | 80.88 ±1.55 | 88.75± 0.52 | 81.48 ± 3.55 |
| CFL SATTLER ET AL. (2021) | 88.30 ± 1.12 | 80.47 ±0.44 | 87.59 ± 0.42 | 82.16 ± 1.73 |
| LOCAL-KMEANS GHOSH ET AL. (2019) | 10.56 ± 1.31 | 10.35 ± 0.71 | 10.00 ±0.20 | 72.89 ± 0.61 |
| IFCA GHOSH ET AL. (2022) | 91.55± 0.81 | **91.80 ± 0.25** | 86.05 ± 0.43 | 85.19 ± 2.01 |

**Simulated Datasets:** We generate clustered FL datasets from MNIST LeCun & Cortes (2010) and CIFAR10 Krizhevsky et al. by splitting them into disjoint sets, one per client. For MNIST, by inverting pixel value, we create 2 clusters (referred to as inverted in table 2) and by rotating the image by 90,180,270 degrees we get 4 clusters. Note that this is a common practice in continual learning Lopez-Paz & Ranzato (2017) and FL Ghosh et al. (2022). We set $m = 100, n = 600$. We obtain Rotated CIFAR10 by creating 2 clusters with the images rotated by 180 degrees. To test with label heterogeneity, we create Label CIFAR10 with 2 clusters. The first cluster contains the first 7 of the 10 labels and the second cluster contains the last 7 of the 10 labels. We set $m = 32$ for both CIFAR10 datasets and $n = 3125$ and $n = 4375$ for Rotated and Label CIFAR10 respectively. To emulate practical FL scenarios, we assume that only a fraction of the nodes participate in the learning procedure. For Rotated and Inverted MNIST, we assume that all the nodes participate, while for Rotated and Label CIFAR10, 50% of the nodes participate. For MNIST, we train a 2-layer feedforward NN, while for CIFAR10, we train a ResNet9 Page (2019). We train Rotated MNIST, Inverted MNIST, Rotated CIFAR10, and Label CIFAR10 for 250, 280, 2400 and 2400 iterations respectively with 2 refine steps for `SR-FCA`.

**Real Datasets:** We use two real federated datasets from leaf database Caldas et al. (2018). We sample $m = 50$ machines from FEMNIST and Shakespeare. FEMNIST is a Federated version of EMNIST with data on each client being handwritten symbols from a different person. Shakespeare is a NLP dataset where the task is next character prediction. For FEMNIST, train a CNN for while for Shakespeare we train a 2-layer stacked LSTM. For clustered FL baselines, we tune $K$, the number of clusters, with $K \in \{2,3,4,5\}$ for FEMNIST and $K \in \{1,2,3,4\}$. We run FEMNIST and Shakespeare for 1000 and 2400 iterations respectively and set number of refine steps to be 1 for `SR-FCA`.

We compare with standard FL baselines – Local (every client trains its own local model) and Global (a single model trained via FedAvg McMahan & Ramage (2017) on all clients). The main baseline we compare to is `IFCA`. Among clustered FL baselines, we consider CFL Sattler et al. (2021), which uses a top-down approach with cosine distance metric, and Local-KMeans Ghosh et al. (2019), which performs KMeans on the model weights of each client's local model. For real datasets, we compare with two additional baselines – FedSoft Ruan & Joe-Wong (2021) and `ONE_SHOT-IFCA` (initial clustering of `IFCA` obtained by `ONE_SHOT`), to assess if these variants can fix the issues of initialization in `IFCA`.

For `SR-FCA`, we tune the parameters $\lambda$ and $\beta$ for trimmed mean and set $t = 2$ and require at most 2 `REFINE` steps. We utilize the following metric based on cross-cluster loss which is better suited to measure distances between clients' models as these are neural networks.

**Definition 5.1** (Cross-Cluster distance)**.** For any two clients $i, j \in [m]$, with corresponding local models $w_i$ and $w_j$ and local empirical losses $f_i$ and $f_j$, we define the cross-cluster loss metric as $\mathsf{dist}_{\text{cross-cluster}}(w_i, w_j) = \frac{1}{2}(f_i(w_j) + f_j(w_i))$.

To extend this definition to distances between cluster $c$ and client $j$, such as those required by `REFINE`, we replace the client model $w_i$ and client loss $f_i$ by the cluster model $w_c$ and empirical cluster loss $f_c$ respectively. Similarly, to obtain distances between clusters $c$ and $c'$, which are required by `MERGE`, we replace the client models and losses with cluster models and losses respectively.

Note that no data is shared between clients to compute the cross-cluster distances, rather the models are shared between clients via the server. To compute the cross-cluster distance between two clients $i$ and $j$, we need to

Table 3: Average Misclustering error of clustered FL algorithms on test set across 5 random seeds for simulated datasets. The lowest error is **bold**. `SR-FCA` is competitve with `IFCA` and beats it for Rotated CIFAR10.

| Baseline | MNIST (INVERTED) | MNIST (ROTATED) | CIFAR (ROTATED) | CIFAR (LABEL) |
|---|---|---|---|---|
| SR-FCA | **0.0** | **0.0** | **0.0** | **0.0** |
| CFL | 0.08 | 0.14 | 0.18 | 0.38 |
| LOCAL-KMeans | 0.36 | 0.28 | 0.38 | 0.26 |
| IFCA | **0.0** | **0.0** | 0.50 | 0.5 |

Table 4: Test Accuracy and standard deviations across 5 seeds on Real datasets. Highest accuracy is **bold**. `SR-FCA` consistently outperforms `IFCA`.

| Baseline | FEMNIST | Shakespeare |
|---|---|---|
| SR-FCA | **83.83**± 1.49 | **48.54 ± 0.69** |
| Local | 66.18 ± 2.14 | 33.86 ±1.22 |
| Global | 80.00± 3.02 | 45.28 ± 0.78 |
| CFL | 79.48 ± 3.48 | 44.14 ± 1.03 |
| IFCA | 81.93± 1.56 | 46.12 ± 1.22 |
| FedSoft | 78.74 ± 2.61 | 46.98 ± 1.25 |
| ONE_SHOT -IFCA | 81.62 ± 2.29 | 45.56 ± 1.15 |

compute $f_j(w_i)$ and $f_i(w_j)$. To compute $f_j(w_i)$, the server sends the model of client $i$, $w_i$, to client $j$ and client $j$ computes its loss on the model $w_i$ and reports it back to the server. $f_i(w_j)$ is computed similarly. To compute the cross-cluster distance between client $i$ and cluster $c$, we need to compute $f_i(w_c)$ and $f_c(w_i)$. $f_i(w_c)$ can be computed by sending the model $w_c$ to client $i$. To compute the quantity $f_c(w_i)$, the server sends the model $w_i$ to each of the clients in cluster $c$ to compute client losses on model $w_i$. The cluster loss is simply the average of the losses of all clients in cluster $c$. We use the same procedure to compute cluster loss, but instead of computing it on a client model, we compute it on the cluster model, to obtain cross-cluster distances between two clusters.

Further, for clustered FL baselines (`IFCA`, CFL, Local-KMeans, FedSoft, `ONE_SHOT-IFCA`) on real datasets, we tune the number of clusters. This is not required in `SR-FCA`. Note that for each baseline and dataset pair, we perform 20 trials of random search for hyperparameter tuning.

As our global model corresponds to FedAvg, for a fair comparison, we use FedAvg McMahan & Ramage (2017) inside the `TrimmedMeanGD` subroutine, bt applying $TrMean_\beta$ operation to the model updates after local steps from different clients. Note that we could have used a different base federated optimization algorithm for global model, for instance, FedProx Li et al. (2020b) or SCAFFOLD Karimireddy et al. (2020). In that case, we would have to modify all our clustered FL algorithms baselines and `SR-FCA`, to use this base federated optimization algorithm for training cluster models.

**Test Metrics:** The test performance of any baseline is obtained by averaging over the clients, the test performance of each client on its model trained by the baseline. For the local baseline, it is the client's local model and for the global baseline it is the single global model. For `SR-FCA` and clustered FL baselines, it is the cluster model for the client. Note that we do not present convergence plots as different algorithms run in different number of stages.

For simulated datasets, the true clustering $\mathcal{C}^\star$ is known, so we report both the test accuracy and misclustering error in table 2 and 3 respectively. For real datasets, the true clustering is not known, therefore, we report only the final test accuracy in table 4. Further, we provide the test accuracy after `ONE_SHOT` and every `REFINE` step in table 5. The total time to run all experiments including hyperparameter tuning on a single NVIDIA-GeForce-RTX-3090 is 2.5 weeks and the code is provid ed here.

## 5.1 Results

Across all datasets, we find that `SR-FCA` is competitive with or outperforms all other algorithms in terms of both misclustering error and test accuracy.

**Comparison with Local and Global:** The Local algorithm has access to little data, while the Global model cannot handle the heterogeneity. Hence, as seen in tables 2 and 4, `SR-FCA` and other clustered FL baselines perform better as they find correct clusters with low heterogeneity inside each cluster.

**Comparison with CFL and Local-KMeans:** CFL and Local-KMeans use the cosine distance between gradients and $l_2$ distance between model weights which are not suitable for NN models. Local-KMeans performs the worst with $\approx 10\%$ test accuracy for simulated datasets. For real datasets, it's accuracy is $\leq 5\%$ so we do not report it. `SR-FCA` and `IFCA` use cross-cluster loss and client loss respectively, which are better suited to NN models, thus outperforming these baselines (see tables 2 and 4)

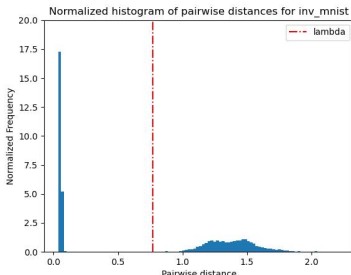 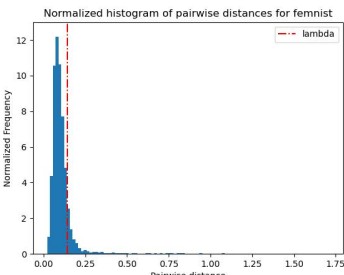

Figure 3: Histogram of pairwise distances after `ONE_SHOT` and $\lambda$ in `SR-FCA` for Inverted MNIST (Left) and FEMNIST(Right).

**Comparison with `IFCA`:** On **simulated datasets** (tables 2 and 3), we find that `IFCA` recovers $\mathcal{C}^\star$ and outperforms `SR-FCA` marginally for MNIST datasets. This is due to MNIST being a simpler and easier to learn dataset, even after adding heterogeneity via rotations or inversions. In contrast, for CIFAR10, the learning task is much more difficult, and `IFCA`, without proper initialization, ends up with all clients in only a single cluster after a few rounds resulting in a misclustering of 0.5, as seen in table 3. Thus it's performance is comparable to the global baseline in terms of test accuracy as seen in table 2. From table 3, we see that `SR-FCA` correctly identifies $\mathcal{C}^\star$ and comprehensively beats `IFCA` in terms of test accuracy in table 2.

On **real datasets**, for `IFCA`, we need to find the correct number of clusters by tuning. For a random sample of clients, the true number of clusters might not be the same. `SR-FCA` can compute both the correct clustering and cluster models for every random sample, allowing it to beat `IFCA`, which fits the same number of clusters to every random sample. The difference is more pronounced for the more difficult Shakespeare dataset than the easier FEMNIST dataset. Further, the variants of FedSoft and `ONE_SHOT-IFCA`, have similar test performance to `IFCA` on real datasets. For FedSoft, which is a soft-clustering version of `IFCA`, the issue of initialization remains unresolved. Running `IFCA` after `ONE_SHOT` can only re-cluster the clients thus results in a similar performance. In short, `SR-FCA` outperforms `IFCA` as well as its variants.

**Intermediate Steps of `SR-FCA`:** For MNIST, multiple `REFINE` steps are necessary for best performance, however, for CIFAR10 and real datasets, only a single `REFINE` step achieves best performance (see table 5).

**Distribution of pairwise distances** Note that $\lambda$ is an important hyperparameter for our algorithm and its choice depends on the distribution of pairwise distances. In Figure 3, we provide the histogram of the pairwise distances after `ONE_SHOT` for Inverted MNIST and FEMNIST respectively. For simulated datasets, there is a clear separation between the pairwise distances – clients in the same clusters have very low pairwise distance and clients in different clusters have very high pairwise distances. Therefore, any choice of $\lambda$ in the middle suffices. However, for real datasets, there isn't a clear separation between clusters. Therefore, we use hyperparameter tuning to choose the which maximizes test accuracy.

## 6 Conclusion

We conclude with a potential scope of improvement in `SR-FCA`. `SR-FCA` trains models obtained via `TrimmedMeanGD()` which assumes $\beta$ fraction of users inside each cluster are corrupted. Instead, if we run any federated optimization algorithm which can accommodate low heterogeneity, for instance FedProx Li et al. (2020b), inside each cluster in the final clustering $\mathcal{C}_R$, then we may obtain an improved error in theory ($\Lambda$ in theorem 4.14). We leave this as a future work.

**Acknowledgements.** This research is supported in part by NSF awards 2112665, 2217058, and 2133484.

Table 5: Test Accuracies of intermediate steps of `SR-FCA` for all datasets

| BASELINE | MNIST (INVERTED) | MNIST (ROTATED) | CIFAR10 (ROTATED) | CIFAR10 (LABEL) | FEMNIST | SHAKESPEARE |
|---|---|---|---|---|---|---|
| AFTER `ONE_SHOT` | $76.52 \pm 0.54$ | $85.55 \pm 0.19$ | $75.87 \pm 0.33$ | $80.09 \pm 0.53$ | $66.18 \pm 2.14$ | $33.86 \pm 1.22$ |
| AFTER $1^{st}$ `REFINE` | $91.88 \pm 0.39$ | $91.63 \pm 0.12$ | $\mathbf{91.38 \pm 0.27}$ | $\mathbf{93.06 \pm 0.20}$ | $\mathbf{83.83 \pm 1.49}$ | $\mathbf{48.54 \pm 0.69}$ |
| AFTER $2^{nd}$ `REFINE` | $\mathbf{92.03 \pm 0.30}$ | $\mathbf{91.66 \pm 0.13}$ | $\mathbf{91.38 \pm 0.27}$ | $\mathbf{93.06 \pm 0.20}$ | $\mathbf{83.83 \pm 1.49}$ | $\mathbf{48.54 \pm 0.69}$ |

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

# Appendix for "An Improved Federated Clustering Algorithm with Model-based Clustering"

## A  Proof of  proposition 4.1

According to the proposition, for two users $i$ and $j$, the data is generated by first sampling each coordinate of $x \in \mathbb{R}^d$ from $\mathcal{N}(0,1)$ iid and then computing $y$ as –

$$y_i = \langle x, w_i^\star \rangle + \epsilon_i$$

where $\epsilon_i \overset{iid}{\sim} \mathcal{N}(0, \sigma^2)$. Then, the distribution of $y_i | x$ is $\mathcal{N}(\langle x, w_i^\star \rangle, \sigma^2)$. Therefore, the $KL$ divergence between $y_i | x$ and $y_j | x$ is given by

$$KL(p(y_i|x) \| p(y_j|x)) = \frac{\langle w_i^\star - w_j^\star, x \rangle^2}{2\sigma^2}$$

Therefore, if we take expectation wrt $x$, we have

$$\mathbb{E}_x[KL(p(y_i|x) \| p(y_j|x))] = \frac{d \| w_i^\star - w_j^\star \|^2}{2\sigma^2}$$

## B  Proof of  lemma 4.6

In `ONE_SHOT()`, $\mathcal{C}_0 = \mathcal{C}^\star$, if all the edges formed in the graph are correct. This means that if $i,j$ are in the same cluster in $\mathcal{C}^\star$, then $\| w_{i,T} - w_{j,T} \| \le \lambda$ and if $i,j$ are in different clusters, $\| w_{i,T} - w_{j,T} \| > \lambda$.

Note that,

$$w_{i,T} - w_{j,T} = (w_i^\star - w_j^\star) + (w_{i,T} - w_i^\star) - (w_{j,T} - w_j^\star)$$

Now, if we apply triangle inequality, we obtain

$$\mathsf{dist}(w_{i,T}, w_{j,T}) \ge \mathsf{dist}(w_i^\star, w_j^\star) - \Xi_{i,j}, \quad \mathsf{dist}(w_{i,T}, w_{j,T}) \le \mathsf{dist}(w_i^\star, w_j^\star) + \Xi_{i,j}$$

where $\Xi_{i,j} = \sum_{k=i,j} \mathsf{dist}(w_{k,T}, w_k^\star)$. This decomposition forms the key motivation for our algorithm.

Therefore, if $i,j$ are in the same cluster, then a sufficient condition for edge $(i,j)$ to be incorrect is

$$\lambda \le \mathsf{dist}(w_i^\star, w_j^\star) + \Xi_{i,j} \implies \Xi_{i,j} \ge \lambda - \epsilon_1$$

Similarly, if $i,j$ are in different clusters, then a sufficient condition for edge $(i,j)$ to be incorrect is

$$\lambda \ge \mathsf{dist}(w_i^\star, w_j^\star) - \Xi_{i,j} \implies \Xi_{i,j} \ge \epsilon_2 - \lambda$$

Therefore, we can set $\Delta_\lambda = \min\{\epsilon_2 - \lambda, \lambda - \epsilon_1\}$, and then a sufficient condition for any edge to be incorrect is $\max_{i,j} \Xi_{i,j} \ge \Delta_\lambda$.

Thus,

$$
\begin{aligned}
\Pr[\mathcal{C}^\star \neq \mathcal{C}_0] &\le \Pr[\text{at least 1 edge is incorrect}] \\
&\le \Pr[\max_{i,j} \Xi_{i,j} \ge \Delta_\lambda] \\
&\le \Pr[\max_{i,j} \sum_{k=i,j} \| w_{k,T} - w_k^\star \| \ge \Delta_\lambda] \\
&\le \Pr[\max_{i,j} \max_{k=i,j} (\| w_{k,T} - w_k^\star \| \ge \frac{\Delta_\lambda}{2}] \\
&\le \Pr[\max_{i \in [m]} \| w_{i,T} - w_i^\star \| \ge \frac{\Delta_\lambda}{2}]
\end{aligned}
\tag{3}
$$

The second and third inequalities are obtained by expanding the terms. The fourth inequality is obtained by $\Pr[a+b \geq c] \leq \Pr[\max\{a,b\} \geq c/2]$. For the fifth inequality, we merge $\max_{i,j} \max_{k=i,j}$ into $\max_{i \in [m]}$. As we can see in Equation equation 3, we need to bound $\|w_{i,T} - w_i^\star\|$ for each node $i$. The subsequent Lemma allow us to bound this quantities.

**Lemma B.1** (Convergence of $w_{i,T}$). *Let $\frac{n^{2/3}\Delta^{4/3}}{D^{2/3}\hat{L}^{2/3}} \lesssim b_1 d$, for some constant $b_1 > 0$. Then, after running* `ONE_SHOT()` *with $\eta \leq \frac{1}{L}$, for some constant $b_2 > 0$, under assumption 4.3 ,assumption 4.4 and assumption 4.5, we have*

$$\Pr[\|w_{i,T} - w_i^\star\| \geq \frac{\epsilon_2 - \epsilon_1}{4}] \leq d \ \exp(-n\frac{b_2\Delta}{\hat{L}\sqrt{d}}),$$

*where $\Delta = \frac{\mu}{2}(\frac{\Delta_\lambda}{2} - (1-\frac{\mu}{L})^{T/2}D)$ and $n = \min_{i \in [m]} n_i$.*

This lemma follows from Yin et al. (2018). The complete proof of this Lemma is present in appendix B.1.

Now, we can apply lemma B.1 in Eq equation 3.

$$\Pr[\mathcal{C}_0 \neq \mathcal{C}^\star] \leq \Pr[\max_{i \in [m]}\|w_{i,T} - w_i^\star\| \geq \frac{\Delta_\lambda}{2}]$$

$$\leq m\max_{i \in [m]}\Pr[\|w_{i,T} - w_i^\star\| \geq \frac{\Delta_\lambda}{2}]$$

$$\leq md \ \exp(-n\frac{b_2\Delta}{\hat{L}\sqrt{d}})$$

For the second inequality, we use $\Pr[\max_{i \in [m]} a_i \geq c] \leq \sum_{i \in [m]}\Pr[a_i \geq c] \leq m\max_{i \in [m]}\Pr[a_i \geq c]$, which follows from union bound.

Note that for $p < 1$, we need the separation to be order of $\Theta(\sqrt{\frac{\log m}{n}})$.

## B.1    Proof of lemma B.1

We utilize results from Yin et al. (2018), which hold for `TrimmedMeanGD` to analyze convergence for a single node as they yield stronger guarantees under the given assumptions.

**Lemma B.2** (Convergence of $w_{i,T}$). *If assumption 4.3,assumption 4.4,and assumption 4.5 hold, and $\eta \leq \frac{1}{L}$, then*

$$\|w_{i,T} - w_i^\star\| \leq (1-\kappa^{-1})^{T/2}D + \frac{2}{\mu}\Lambda_i \quad \forall i \in [m] \tag{4}$$

*where $\kappa = \frac{L}{\mu}$ and $\Lambda_i$ is a positive random variable with*

$$\Pr[\Lambda_i \geq \sqrt{2d}r + 2\sqrt{2}\delta\hat{L}] \leq 2d(1+\frac{D}{\delta})^d\exp(-n\min\{\frac{r}{2\hat{L}}, \frac{r^2}{2\hat{L}^2}\}) \tag{5}$$

*for some $r, \delta > 0$.*

We provide the proof of this lemma in appendix C.8.

Using the above Lemma, we can bound the probability $\Pr[\|w_{i,T} - w_i^\star\| \geq \frac{\Delta_\lambda}{2}]$

$$\Pr[\|w_{i,T} - w_i^\star\| \geq \frac{\Delta_\lambda}{2}] \leq \Pr[2(1-\kappa^{-1})^{T/2}D + \frac{2}{\mu}\Lambda_i + \geq \frac{\Delta_\lambda}{2}]$$

$$\leq \Pr[\Lambda_i \geq \Delta], \quad \text{where } \Delta = \frac{\mu}{2}(\frac{\Delta_\lambda}{2} - (1-\kappa^{-1})^{T/2}D)$$

$$\leq \Pr[\sqrt{2d}r + 2\sqrt{2}\delta\hat{L} \geq \Delta]$$

$$\leq d\exp(-nb_2\frac{\Delta}{\hat{L}\sqrt{d}})$$

for some constants $b_1, b_2, b_3, b_4 > 0$, where we set $r = b_3\hat{L}\max\{\frac{\Delta}{\hat{L}\sqrt{d}}, \sqrt{\frac{\Delta}{\hat{L}\sqrt{d}}}\}$ and $\delta = b_4\frac{\Delta}{\hat{L}}$, and for $b_1 d \leq \frac{n^{2/3}\Delta^{4/3}}{D^{2/3}\hat{L}^{4/3}}$, such that $\sqrt{2d}r + 2\sqrt{2}\delta\hat{L} \geq \Delta$ and $n\min\{\frac{r}{2\hat{L}}, \frac{r^2}{2\hat{L}^2}\} > \frac{Dd}{\delta}$ in lemma B.2.

# C  Proof of theorem 4.8

## C.1  Preliminaries

First, we define certain random variables and their respective probabilities which we will use throughout this proof. Since the edge based analysis and corresponding clique identification involves a lot of dependent events, we try to decompose the absence/presence of edge into a combination of independent events.

Define,

$$X_{ij} = \begin{cases} 1 & \text{If the edge } (i,j) \text{ in } \mathcal{C}_0 \text{ is incorrect in } \mathcal{C}^\star \\ 0 & \text{Otherwise} \end{cases} \tag{6}$$

An edge $(i,j)$ in $\mathcal{C}_0$ is incorrect in $\mathcal{C}^\star$ if either it is present in $\mathcal{C}^\star$ and absent in $\mathcal{C}_0$ or vice versa. We analyze the probability of this event for the case when $\mathcal{C}^\star$ contains the edge $(i,j)$. The case when $\mathcal{C}^\star$ doesn't contain edge $(i,j)$ and it is present in $\mathcal{C}_0$ has exaclty same probability. When $\left\| w_i^\star - w_j^\star \right\| \leq \epsilon_1$, then edge is present is $\mathcal{C}^\star$. If it is absent in $\mathcal{C}_0$, then

$$\Pr[X_{ij} = 1] \leq \Pr[\Xi_{i,j} \geq \Delta_\lambda]$$
$$\leq \Pr[\Lambda_i + \Lambda_j \geq 2\Delta]$$

The analysis is similar to the proof of `ONE_SHOT()` in appendix B.

Note that the random variables $\{X_{ij}\}$ are not independent. We now define independent random variables $X_i$ such that

$$X_i = \begin{cases} 1 & \text{If } \Lambda_i \geq \Delta \\ 0 & \text{Otherwise} \end{cases} \tag{7}$$

Thus, we can see that $X_{ij} \leq X_i + X_j$. Additionally,

$$\Pr[X_i = 1] \leq \Pr[\Lambda_i \geq \Delta] \leq \frac{p}{m} \tag{8}$$

This follows from analysis of `ONE_SHOT()` in appendix B.

We can further generalize this notion to the random variables defined as $Y_{i,\gamma}$.

$$Y_{i,\gamma} = \begin{cases} 1 & \text{If } \Lambda_i \geq \gamma\Delta, \gamma \in (0,2) \\ 0 & \text{Otherwise} \end{cases} \tag{9}$$

Then,

$$\Pr[Y_{i,\gamma} = 1] \leq \Pr[\Lambda_i \geq \gamma\Delta] \leq d\exp(-nb_2 \frac{\gamma\Delta}{\hat{L}\sqrt{d}}) = (\frac{p}{m})^\gamma$$

Note that the set of random variables $\{Y_{i,\gamma}\}_{i=1}^m$ are mutually independent random variables.

Further, we define the $\omega_c^\star$ for every cluster $c \in \mathrm{rg}(\mathcal{C}_0)$. Let $c' \in \mathcal{C}^\star$ be the cluster label of node $c$. If $G_c = \{i : i \in [m], \mathcal{C}^\star(i) = c'\}$, which is the set of nodes in $c$ which were from $c'$ in the original clustering, then we can define $\omega_c^\star$ and $F_c(w)$ as

$$\omega_c^\star = \underset{w \in \mathcal{W}}{\arg\min} \mathbb{E}\Big[\frac{1}{|G_{c'}|} \sum_{i \in G_{c'}} f_i(w)\Big] \tag{10}$$

$$= \underset{w \in \mathcal{W}}{\arg\min} \frac{1}{|G_{c'}|} \sum_{i \in G_{c'}} F_i(w) = \underset{w \in \mathcal{W}}{\arg\min} F_c(w) \tag{11}$$

We use this definition of $\omega_c^\star$ in the appendices C.5 and C.6.

### C.2 Analysis of `REFINE()`

Our goal is to compute total probability of error for `REFINE()` to fail. If we define this error as $\mathcal{C}_1 \neq \mathcal{C}^\star$, then we can define the main sources of error for this event.

1. $\exists c \in \mathrm{rg}(\mathcal{C}^\star)$ **such that no cluster in** $\mathcal{C}_0$ **has cluster label** $c$: If the a cluster $c \in \mathrm{rg}(\mathcal{C}^\star)$ is absent in $\mathcal{C}_0$, then subsequent steps of `REFINE()` will never be able to recover it, as they only involve node reclustering and merging existing clusters. The lemma presented below gives an upper bound on the probability of this event.

   **Lemma C.1.** *Under the conditions of lemma 4.6 and if* $t = \Theta(c_{\min})$, *then there exists constant* $a_1 > 0$ *such that*

   $$\Pr[\exists c \in \mathrm{rg}(\mathcal{C}^\star) \text{ such that no cluster in } \mathcal{C}_0 \text{ has cluster label } c]$$
   $$\leq \frac{m}{c_{\min}}\exp(-a_1 c_{\min})$$

   The proof of this Lemma is presented in appendix C.3

2. **Each cluster** $c \in \mathrm{rg}(\mathcal{C})_0$ **should have** $< \alpha$ **fraction of impurities for some** $\frac{1}{2} > \beta > \alpha$: If some cluster has more than $\alpha$-fraction of impure nodes, then we cannot expect convergence guarantees for `TrimmedMeanGD`$_\beta$.

   The below lemma bounds the probability of this error as

   **Lemma C.2.** . *For some constants* $0 < \alpha < \beta < \frac{1}{2}, a_2 \geq 0, \gamma_1 \in (1,2)$ *and* $\alpha t = \Theta(m)$, *under the conditions in lemma 4.6, we have*

   $$\Pr[\exists c \in \mathrm{rg}(\mathcal{C}_0) \text{ which has } > \alpha \text{ fraction of impurities }]$$
   $$\leq \frac{m}{t}\exp(-a_2 m) + (1-\alpha)m(\frac{p}{m})^{\gamma_1}$$

   The proof of this Lemma is presented in appendix C.4.

3. `MERGE()` **error:** We define this as the error for the `MERGE()` to fail. Even though `MERGE()` operates after `RECLUSTER()`, `RECLUSTER()` does not change the cluster iterates. The goal of `MERGE()` is to ensure that all clusters in $\mathcal{C}_0$ with the same cluster labels are merged. Therefore, we define `MERGE()` error as the event when either two clusters with same cluster label are not merged or two clusters with different cluster labels are merged. The below lemma bounds this probability.

   **Lemma C.3.** *If* $\min\{\frac{n^{2/3}\Delta^{4/3}}{D^{2/3}\hat{L}^{2/3}}, \frac{n^2\Delta'^2}{\hat{L}^2\log(c_{\min})}\} \geq u_1 d$ *for some constants* $u_1 > 0$, *then for some constant* $a_3' > 0$, *where* $\Delta' = \Delta - \frac{\mu B}{2} > 0$, *where* $B = \sqrt{\frac{2\hat{L}\epsilon_1}{\mu}}$, *we have*

   $$\Pr[\textit{MERGE() Error}] \leq \frac{4dm}{t}\exp(-a_3' n\frac{\Delta'}{2\hat{L}})$$

   The proof of this Lemma is presented in appendix C.5.

4. `RECLUSTER()` **error:** This event is defined as a node going to the wrong cluster after both `MERGE()` and `REFINE()` operations. After `MERGE()`, each cluster in $\mathcal{C}_0$ corresponds to a single cluster in $\mathcal{C}_1$. Therefore, we incur an error due to the `RECLUSTER()` operation if any node $i$ does not go to the cluster $c \in \mathcal{C}_1$ which has cluster label $\mathcal{C}^\star(i)$. The below lemma provides an upper bound on the probability of this error.

   **Lemma C.4.** *If* $\min\{\frac{n^{2/3}\Delta^{4/3}}{D^{2/3}\hat{L}^{2/3}}, \frac{n^2\Delta'^2}{\hat{L}^2\log(c_{\min})}\} \geq u_2 d$ *for some constants* $u_2 > 0$, *then for some constants* $a_3'' > 0$ *and* $\gamma_2 \in (1, 2 - \frac{\mu B}{2\Delta})$, *we have*

   $$\Pr[\textit{RECLUSTER()error}] \leq 4d\frac{m}{t}\exp(-a_3'' n\frac{\Delta'}{2\hat{L}}) + m(\frac{p}{m})^{\gamma_2} \tag{12}$$

The proof of this Lemma is presented in appendix C.6.

The total probability of error after for a single step of `REFINE()` is the sum of probability of errors for these 4 events by the union bound. Therefore,

$$\Pr[\mathcal{C}_1 \neq \mathcal{C}^\star] \leq \frac{m}{c_{\min}} \exp(-a_1 c_{\min}) + \frac{m}{t} \exp(-a_2 m)$$

$$+ (1-\beta)m\left(\frac{p}{m}\right)^{\gamma_1} + 8d\frac{m}{t}\exp\left(-a_3 n\frac{\Delta'}{2\hat{L}}\right) + m\left(\frac{p}{m}\right)^{\gamma_2}$$

where we set $a_3 = \min\{a_3', a_3''\}$ .

For some small constants $\rho_1 > 0$, $\rho_2 \in (0,1)$, we can choose $\gamma_1 \in (1,2)$, $\beta \in (0, \frac{1}{2})$ and $\gamma_2 \in (1, 2 - \frac{\mu B}{2\Delta})$ such that $(1-\beta)(\frac{p}{m})^{\gamma_1 - 1} + (\frac{p}{m})^{\gamma_2 - 1} \leq \frac{\rho_1}{2m^{1-\rho_2}}$ and for large enough $m$, $\Delta'$ and $n$, $\frac{m}{c_{\min}}\exp(-a_1 c_{\min}) + \frac{m}{t}\exp(-a_2 m) + 8d\frac{m}{t}\exp(-a_3 n\frac{\Delta'}{2\hat{L}}) \leq \frac{\rho_1}{2m^{1-\rho_2}}p$. This happens because we have terms of $\exp(-m)$, $\exp(-c_{\min})$ and $\exp(-n\Delta')$, which decrease much faster than $\frac{p}{m}$ which has terms of $\mathcal{O}(m\exp(-n\Delta))$, where $\Delta$ and $\Delta'$ are of the same order. Therefore, the total probability of error can be bounded by

$$\Pr[\mathcal{C}_1 \neq \mathcal{C}^\star] \leq \frac{\rho_1}{m^{1-\rho_2}}p \tag{13}$$

## C.3  Proof of lemma C.1

$$\Pr[\exists c \in \mathrm{rg}(\mathcal{C}^\star) \text{ such that no cluster in } \mathcal{C}_0 \text{ has cluster label } c]$$
$$\leq \sum_{c \in \mathcal{C}^\star} \Pr[\text{No cluster in } \mathcal{C}_0 \text{ has cluster label } c] \tag{14}$$

Here, we use union bound over the clusters for the second inequality. Now, we analyze the probability that no cluster in $\mathrm{rg}(\mathcal{C}_0)$ has cluster label $c$ for some $c \in \mathrm{rg}(\mathcal{C}^\star)$. Consider a cluster in $\mathrm{rg}(\mathcal{C}_0)$. This cluster has cluster label $c$ if a majority of its nodes are from cluster $c \in \mathrm{rg}(\mathcal{C}^\star)$. Since the size of each cluster in $\mathrm{rg}(\mathcal{C}_0)$ is atleast $t$ and there are $K$ clusters in $\mathrm{rg}(\mathcal{C}^\star)$, if all clusters in $\mathrm{rg}(\mathcal{C}_0)$ have $\leq \frac{t}{K}$ nodes from cluster $c$, then no cluster will have cluster label $c$.

Assume that the clique formed by nodes from cluster $c$ has $r$ nodes. Then, every node $i$ in cluster $c$, must have $S_c - r$ edges absent, which correspond to the edges between a node of the clique and those outside it. Thus, we obtain,

$$\Pr[\text{No cluster in } \mathcal{C}_0 \text{ has cluster label } c] \leq \Pr\left[\bigcap_{\mathcal{C}^\star(i)=c}\left\{\sum_{j \neq i, \mathcal{C}^\star(i)=c} X_{ij} > S_c - \frac{t}{K}\right\}\right]$$

$$\leq \Pr\left[\sum_{\mathcal{C}^\star(i)=\mathcal{C}^\star(j)=c} X_{ij} > S_c\left(S_c - \frac{t}{K}\right)\right]$$

$$\leq \Pr\left[\sum_{\mathcal{C}^\star(i)=\mathcal{C}^\star(j)=c} (X_i + X_j) > S_c\left(S_c - \frac{t}{K}\right)\right]$$

$$\leq \Pr\left[\frac{1}{S_c}\sum_{\mathcal{C}^\star(i)=c} X_i > 1 - \frac{t}{KS_c}\right]$$

$$\leq \exp\left(-\left(1 - \frac{t}{KS_c} - \frac{p}{m}\right)^2 S_c\right)$$

$$\leq \exp(-a_1 c_{\min})$$

In the first step, we require each node $i$ to have $S_c - \frac{t}{K}$ wrong edges. For the second inequality, we remove the intersection and thus, the total number of incorrect edges has to be $S_c(S_c - \frac{t}{K})$, since each node has $S_c - \frac{t}{K}$ incorrect edges. For the third inequality, we use $X_{ij} \leq X_i + X_j$ and collect the terms of $X_i$ for the fourth inequality. In the fifth inequality, we obtain a condition on the sum of independent Bernoulli random variables each with mean $\frac{p}{m}$. Therefore, we can apply Chernoff bound for their sum to obtain the fifth inequality.

A necessary condition for us is $1-\frac{t}{KS_c}-\frac{p}{m}>0$ which translates to $t<KS_c(1-\frac{p}{m})$. If we select $t\leq c_{min}-1$, this inequality is always satisfied. Note that we want the term $\left(1-\frac{t}{KS_c}-\frac{p}{m}\right)^2>a_1$, for some positive constant $a_1$. If we choose $t=\Theta(m)$, which is possible if $t=\Theta(c_{\min})$ as we assume $c_{\min}=\Theta(m)$, then this is satisfied. We use the lower bound $a_1$ and $S_c\geq c_{\min}$ to obtain the final inequality. Plugging this in Eq equation 14, we obtain our result.

### C.4 Proof of lemma C.2

$$\Pr[\exists c\in\mathrm{rg}(\mathcal{C}_0)\text{ which has }\geq\alpha\text{ fraction of impurities}]$$
$$\leq\sum_{c\in\mathrm{rg}(\mathcal{C}_0)}\Pr[\text{cluster }c\text{ has }\geq\alpha\text{ fraction of wrong nodes}]\tag{15}$$

We use a simple union bound on clusters in $\mathcal{C}_0$ for the above inequality. Let the set of nodes in the cluster $c$ which are from same cluster of $\mathcal{C}^\star$ as the cluster label of $c$, i.e., which are not impurities, be $R_c$. Then let $Q_c=|R_c|$. Let $Q_c'$ denote the number of impurities in cluster $c$.

$$\Pr[\text{cluster }c\text{ has }\geq\alpha\text{ fraction of wrong nodes}]\leq\Pr[Q_c'\geq\frac{\alpha}{1-\alpha}Q_c]$$
$$\Pr[Q_c'\geq\alpha t]$$

We use the fact that $Q_c+Q_c'\geq t$, which is the minimum size of any cluster, for the second inequality.

Now, we analyze the probability of a single node to be incorrect. A node is an impurity in cluster $c$ if it has an edge to each of nodes in $R_c$.

$$\Pr[\text{Node }i\text{ is an impurity in cluster c}]\leq\Pr[\min_{j\in R_c}\|w_{i,T}-w_{j,T}\|\leq\lambda]\tag{16}$$
$$\leq\Pr[\min_{j\in R_c}(\|w_i^\star-w_j^\star\|-\Xi_{i,j})\leq\lambda]$$
$$\leq\Pr[\Lambda_i+\max_{j\in R_c}\Lambda_j\geq2\Delta]$$

Now, if $\max_{j\in R_c}\Lambda_j\leq\gamma_1\Delta$, for $\gamma_1\in(1,2)$, then we need $\Lambda_i\geq(2-\gamma_1)\Delta$ for error.

Using the definition of random variables in appendix C.1

$$\Pr[Q_c'\geq\alpha t]\leq\Pr[Q_c'\geq\alpha t|\max_{j\in R_c}\Lambda_j\leq\gamma_1\Delta]+\Pr[\max_{j\in R_c}\Lambda_j\geq\gamma_1\Delta]$$
$$\leq\Pr[\sum_{i=1}^m Y_{i,2-\gamma_1}\geq\alpha t]+\Pr[\max_{j\in R_c}\Lambda_j\geq\gamma_1\Delta]$$

For the first inequality, we use union bound over the value of $\max_{j\in R_c}\Lambda_j$ and for the second inequality, we need atleast $\alpha t$ impurities, so atleast $\alpha t$ of all $Y_{i,2-\gamma_1}$ should be 1.

We now bound the two terms in the final inequality separately.

For the second term, if $\max_{j\in R_c}\Lambda_j\geq\gamma_1\Delta$.

$$\Pr[\max_{j\in R_c}\Lambda_j\geq\gamma_1\Delta]\leq Q_c\Pr[Y_{j,\gamma_1}=1]\leq Q_c(\frac{p}{m})^{\gamma_1}$$

Here, we use union bound over all elements in $R_c$ for the first inequality and the second inequality is plugging in the value of $\Pr[Y_{j,\gamma_1}=1]$, which we have already computed.

Now, we need to provide a bound on $Q_c$. Note that if $Q_c$ denotes the correct number of nodes, which corresponds to the majority of nodes, then $Q_c\leq(1-\alpha)S_c$, where $S_c$ is the size of the cluster $c$.

For the first term, we can use Chernoff bound as $Y_{i,2-\gamma_1}$ are independent random variables with expectation $\frac{p}{m}$

$$\Pr[\frac{1}{m}\sum_{i=1}^m Y_{i,2-\gamma_1}\geq\alpha\frac{t}{m}]\leq\exp(-(\alpha\frac{t}{m}-\mathbb{E}[Y_{i,2-\gamma_1}])^2 m)\leq\exp(-a_2 m)$$

We need $\alpha\frac{t}{m} \geq \mathbb{E}[Y_{i,2-\gamma_1}]$, which implies $\alpha t \geq 1$, since $Y_{i,2-\gamma_1}$ is a bernoulli random variable. Further, we require $\alpha t = \Theta(m)$, so that we can bound the probability using a constant $a_2 \geq 0$. If we choose $\gamma_1$ as a constant independent of $m$, then we are done.

Now, plugging all these inequalities into Eq equation 15, we get

$$\Pr[\exists c \in \mathrm{rg}(\mathcal{C}_0) \text{ which has } \geq \alpha \text{ fraction of wrong nodes}]$$
$$\leq \mathrm{rg}(\mathcal{C}_0)\exp(-a_2 m) + \sum_{c \in \mathrm{rg}(\mathcal{C}_0)} (1-\alpha)S_c(\frac{p}{m})^{\gamma_1}$$
$$\leq |\mathrm{rg}(\mathcal{C}_0)|\exp(-a_2 m) + (1-\alpha)m(\frac{p}{m})^{\gamma_1}$$
$$\leq \frac{m}{t}\exp(-a_2 m) + (1-\alpha)m(\frac{p}{m})^{\gamma_1}$$

For the second inequality, we use $\sum_{c \in \mathcal{C}_0} S_c = m$ and for the third inequality, we use $|\mathrm{rg}(\mathcal{C}_0)|t \leq m$.

## C.5  Proof of lemma C.3

First, let $i,j \in [m]$ be a node in cluster $c,c' \in \mathrm{rg}(\mathcal{C}_0)$ respectively such that $\mathcal{C}^\star(j)$ and $\mathcal{C}^\star(i)$ are the cluster labels of clusters $c$ and $c'$ respectively. Then, if we repeat our thresholding analysis for $\mathtt{MERGE()}$ operation, we obtain

$$\mathsf{dist}(w_i^\star, w_j^\star) - \Psi_{c,c'} \leq \mathsf{dist}(\omega_{c,T}, \omega_{c',T}) \leq \mathsf{dist}(w_i^\star, w_j^\star) + \Psi_{c,c'}$$
$$\text{where } \Psi_{c,c'} = \mathsf{dist}(\omega_c^\star, w_i^\star) + \mathsf{dist}(\omega_{c'}^\star, w_j^\star) + \sum_{k=c,c'} \mathsf{dist}(w_{k,T}, w_k^\star)$$

We obtain the above equations by a simple application of triangle inequality. Here, $\omega_c^\star$ is as defined in appendix C.1.

To analyze the above quantities, we need to bound $\|\omega_c^\star - \omega_{c,T}\|$ and $\|\omega_c^\star - w_j^\star\|$ for some $j \in G_c$. The following Lemmas provide these bounds.

**Lemma C.5** (Convergence of $\omega_{c,T}$). *If assumption 4.3, assumption 4.4 and assumption 4.5 hold, and $\eta \leq \frac{1}{L}$, then*

$$\|\omega_{c,T} - \omega_c^\star\| \leq (1-\kappa^{-1})^{T/2}D + \frac{2}{\mu}\Lambda_c \quad \forall c \in \mathrm{rg}(\mathcal{C}_0) \tag{17}$$

*where $\kappa = \frac{L}{\mu}$ and $\Lambda_c$ is a positive random variable with*

$$\Pr[\Lambda_c \geq \sqrt{2d}\frac{r+3\beta s}{1-2\beta} + \sqrt{2}\frac{2(1+3\beta)}{1-2\beta}\delta\hat{L}]$$
$$\leq 2d(1+\frac{D}{\delta})^d(\exp(-(1-\alpha)S_c n\min\{\frac{r}{2\hat{L}}, \frac{r^2}{2\hat{L}^2}\}) \tag{18}$$
$$+ (1-\alpha)S_c\exp(-n\min\{\frac{s}{2\hat{L}}, \frac{s^2}{2\hat{L}^2}\}))$$

*for some $r,s,\delta > 0$ where $S_c$ is the size of cluster $c$.*

Proof is presented in appendix C.7

**Lemma C.6** (Distance between cluster minima and node minima). *If assumption 4.3 and assumption 4.5 are satisfied then, for all $j \in [m]$, where $j$ is a node in cluster $c \in \mathcal{C}_0$ where $\mathcal{C}^\star(j)$ is the cluster label of node c, we have*

$$\|\omega_c^\star - w_j^\star\| \leq \sqrt{\frac{2\hat{L}\epsilon_1}{\mu}} := B \tag{19}$$

Proof is presented in appendix C.9.

Now, that we have our required quantities, we are ready to analyze the probability of error after the merge and reclustering operations.

First, we analyze the probabilty of $\texttt{MERGE()}$ operation. Note that if correct nodes of $c$ and $c'$ were from the same cluster $\mathcal{C}^\star$ then, $\|w_i^\star - w_j^\star\| \le \epsilon_1, \forall i \in G_c, j \in G_{c'}$. If correct nodes of $c'$ and $c$ were from different clusters in $\mathcal{C}^\star$, then, $\|w_i^\star - w_j^\star\| \ge \epsilon_2, \forall i \in G_c, j \in G_{c'}$. Therefore, the probability of $\texttt{MERGE()}$ error is upper bounded by

$$\Pr[\texttt{MERGE() Error}] \le \Pr[\text{at least 1 edge is incorrect}]$$

$$\le \Pr[\max_{c,c'} \Psi_{c,c'} \ge \Delta_\lambda]$$

$$\le \Pr[\max_{c,c'} \sum_{k=c,c'} \frac{2\Lambda_k}{\mu} \ge \Delta_\lambda - 2(1-\kappa^{-1})^{T/2}D - 2B]$$

$$\le \max_{c \in \text{rg}(\mathcal{C}_0)} \Pr[\Lambda_c \ge \frac{\mu}{2}(\frac{\Delta_\lambda}{2} - (1-\kappa^{-1})^{T/2}D - B)]$$

$$\le \max_{c \in \text{rg}(\mathcal{C}_0)} \Pr[\Lambda_c \ge \Delta'] \tag{20}$$

$$\le \max_{c \in \text{rg}(\mathcal{C}_0)} 4d\exp(-a_3' n \frac{\Delta'}{2\hat{L}})$$

$$\le \sum_{c \in \text{rg}(\mathcal{C}_0)} 4d\exp(-a_3' n \frac{\Delta'}{2\hat{L}}) \le \frac{4dm}{t}\exp(-a_3' n \frac{\Delta'}{2\hat{L}}) \tag{21}$$

For the second inequality, we expand all the terms of $\Phi_{c,c'}$. We set $\Delta' = \frac{\mu}{2}(\frac{\Delta_\lambda}{2} - (1-\kappa^{-1})^{T/2}D - B)$. Then, we set $r = \Theta(\hat{L}\max\{\frac{\Delta'}{S_c\sqrt{d}\hat{L}}, \sqrt{\frac{\Delta'}{S_c\sqrt{d}\hat{L}}}\}), s = \Theta(\hat{L}\max\{\frac{\Delta'}{S_c\sqrt{d}\hat{L}} + \frac{2\log(S_c)}{n}, \sqrt{\frac{\Delta'}{S_c\sqrt{d}\hat{L}} + \frac{2\log(S_c)}{n}}\})$ and $\delta = \Theta(\frac{Dd^{3/2}\hat{L}}{n\Delta'})$. Now, if $d = \Omega(\min\{\frac{n^{2/3}\Delta'^{4/3}}{D^{2/3}\hat{L}^{2/3}}, \frac{n^2\Delta'^2}{\hat{L}^2\log(c_{\min})}\})$, such that $\sqrt{2d}\frac{r+3\beta s}{1-2\beta} + \sqrt{2}\frac{2(1+3\beta)}{1-2\beta}\delta\hat{L} \ge \Delta'$, then there exist some constant $a_3' > 0$ such that the second inequality is satisfied by lemma C.5. We then use the union bound, followed by $|\text{rg}(\mathcal{C}_0)| \le \frac{m}{t}$.

## C.6  Proof of lemma C.4

We can apply our thresholding analysis to

$\|\omega_{c,T} - w_{i,T}\|$ for $c \in \text{rg}(\mathcal{C}_0)$. First, let $j$ be a node in cluster $c$ such that $\mathcal{C}^\star(j)$ is the cluster label of $c$.

$$\text{dist}(w_j^\star, w_i^\star) + \Phi_{c,i} \le \text{dist}(\omega_{c,T}, w_{i,T}) \le \text{dist}(w_j^\star, w_i^\star) + \Phi_{c,i}$$
$$\text{where } \Phi_{c,i} = \text{dist}(\omega_{c,T}, \omega_c^\star) + \text{dist}(\omega_c^\star, w_j^\star) + \text{dist}(w_{i,T}, w_i^\star)$$

From appendix B and appendix C.5, we have bounds for all the terms involved. Note that after merging, each cluster in $\mathcal{C}^\star$ should have only 1 cluster in $\mathcal{C}_1$. Therefore, after we recluster according to $\|\omega_{c,T} - w_{i,T}\|$, we incur an error if $i$ goes to the wrong cluster. Suppose that the $c$ corresponds to the correct cluster for $i$ and $c'$ is the cluster to which it is assigned , with $c, c' \in \text{rg}(\mathcal{C}_1), c \ne c'$. Then,

$$\Pr[\text{Reclustering Error}] \le \Pr[\max_{i \in [m]} \max_{c' \ne c} \|\omega_{c',T} - w_{i,T}\| \le \|\omega_{c,T} - w_{i,T}\|]$$

$$\le \Pr[\max_{i \in [m]} \max_{c' \ne c} \epsilon_2 - \Phi_{c',i} \le \epsilon_1 + \Phi_{c,i}]$$

$$\le \Pr[\max_{i \in [m]} \max_{c' \in \mathcal{C}_0'} \Phi_{c,i} \ge \frac{\epsilon_2 - \epsilon_1}{2}]$$

$$\le \Pr[\max_{i \in [m]} \max_{c' \in \mathcal{C}_0'} (\Lambda_c + \Lambda_i) \ge \Delta + \Delta'] \tag{22}$$

$$\le \Pr[\max_{c \in \mathcal{C}_0'} \Lambda_c \ge \Delta' - (\gamma_2 - 1)\Delta] + \Pr[\max_{i \in [m]} \Lambda_i \ge \gamma_2 \Delta]$$

$$\le \max_{c \in \text{rg}(\mathcal{C}_0)'} \Pr[\Lambda_c \ge \Delta''] + \max_{i \in m} \Pr[\Lambda_i \ge \gamma_2 \Delta] \tag{23}$$

For the second inequality, we use the thresholding analysis on $\|\omega_{c,T} - w_{i,T}\|$. For the third inequality, we rearrange the terms and combine max over $c' \neq c$ with $c$, and use. For the fourth inequality, we expand the terms of $\Phi_{c,T}$ and substitute the values of $\Delta$ and $\Delta'$, using the inequality $\Delta_\lambda \leq \frac{\epsilon_2 - \epsilon_1}{2}$. For the fifth inequality, we use consider some $\gamma_2 \in (1, 2 - \frac{\mu B}{2\Delta})$ and break the terms using union bound such that $\Delta'' = \Delta' - (\gamma_2 - 1)\Delta \geq 0$. Finally, we use the union bound on $c \in \mathrm{rg}(\mathcal{C}_0)'$ and $i \in [m]$.

Now, we bound the two terms in Eq equation 23 separately. The second term can be bounded in terms of $Y_{i,\gamma_2}$. Thus,

$$\max_{i \in [m]} \Pr[\Lambda_i \geq \gamma_2 \Delta] = \max_{i \in [m]} \Pr[Y_{i,\gamma_2} = 1] \leq m(\frac{p}{m})^{\gamma_2} \tag{24}$$

We use expectation of $Y_{i,\gamma_2}$ calculated in appendix C.4 and then bound max by sum.

For the first term, our analysis is similar to that of `MERGE()` error. Assume that there is some constant $u_2 > 1$ such that $\Delta'' \geq u_2 \Delta'$. We set $\delta = \Theta(\frac{Dd^{3/2}\hat{L}}{n\Delta'})$, $r = \Theta(\hat{L} \max\{\frac{\Delta'}{S_c \sqrt{d}\hat{L}}, \sqrt{\frac{\Delta'}{S_c \sqrt{d}\hat{L}}}\})$, $s = \Theta(\hat{L} \max\{\frac{\Delta'}{S_c \sqrt{d}\hat{L}} + \frac{2\log(S_c)}{n}, \sqrt{\frac{\Delta'}{S_c \sqrt{d}\hat{L}} + \frac{2\log(S_c)}{n}}\})$, and if $d = \Omega(\min\{\frac{n^{2/3}\Delta^{4/3}}{D^{2/3}\hat{L}^{2/3}}, \frac{n^2\Delta'^2}{\hat{L}^2 \log(c_{\min})}\})$, such that $\sqrt{2d}\frac{r+3\beta s}{1-2\beta} + \sqrt{2}\frac{2(1+3\beta)}{1-2\beta}\delta\hat{L} \geq \Delta'$, then there exist some constant $a_3'' > 0$ such that the second inequality is satisfied by lemma C.5. We then use the union bound, followed by $|\mathrm{rg}(\mathcal{C}_0)| \leq \frac{m}{t}$.

$$\max_{c \in \mathrm{rg}(\mathcal{C}_0)'} \Pr[\Lambda_c \geq \Delta''] \leq \max_{c \in \mathrm{rg}(\mathcal{C}_0)'} 4d \exp(-a_3'' n \frac{\Delta'}{2\hat{L}}) \tag{25}$$

$$\leq \sum_{c \in \mathrm{rg}(\mathcal{C}_0)'} 4d \exp(-a_3'' n \frac{\Delta'}{2\hat{L}}) \tag{26}$$

$$\leq \frac{4dm}{t} \exp(-a_3'' n \frac{\Delta'}{2\hat{L}}) \tag{27}$$

## C.7 Proof of lemma C.5

First, we use an intermediate Lemma from Yin et al. (2018). This characterizes the behavior of $TrimmedMean_\beta$ gradient estimator.

**Lemma C.7** (TrimmedMean Estimator Variance). *Let $g_c(w)$ be the output of $\mathrm{TrMean}_\beta$ estimator for cluster $c \in \mathcal{C}_0$ with size of cluster $S_c$. If assumption 4.5 holds, then*

$$\|g_c(w) - \nabla F_c(w)\| \leq \Lambda$$
$$where \Pr[\Lambda \geq \sqrt{2d}\frac{r+3\beta s}{1-2\beta} + \sqrt{2}\frac{2(1+3\beta)}{1-2\beta}\delta\hat{L}]$$
$$\leq 2d(1+\frac{D}{\delta})^d \left( \exp(-(1-\alpha)S_c n \min\{\frac{r}{2\hat{L}}, \frac{r^2}{2\hat{L}^2}\}) \right.$$
$$\left. + (1-\alpha)S_c \exp(-n\min\{\frac{s}{2\hat{L}}, \frac{s^2}{2\hat{L}^2}\}) \right) \tag{28}$$

*for some $r, s, \delta > 0$.*

*Proof.* The proof of this Lemma follows from coordinate-wise sub-exponential distribution of $\nabla F_c$. Since loss per sample $f(w,z)$ is Lipschitz in each of its coordinates with Lipschitz constant $L_k$ for $k \in [d]$. Thus, $F_c(w)$ is also $L_k$-Lipschitz for each coordinate $k \in [d]$ from corollary E.6. Now, every subgaussian variable with variance $\sigma^2$ is $\sigma$-sub exponential. Thus, each coordinate of $\nabla_w f(w,z)$ is $\hat{L}$-sub-exponential, since $\hat{L} > L_k, \forall k \in [d]$. The remainder of proof can be found in Appendix E.1 in Yin et al. (2018). $\qquad\square$

Now, using the above Lemma, we can bound the iterate error for a cluster $c \in \mathcal{C}_0$. Consider $\|\omega_{c,t+1} - \omega_c^\star\|^2$,

$$
\begin{aligned}
\|\omega_{c,t+1} - \omega_c^\star\| &\leq \|proj_{\mathcal{W}}\{\omega_{c,t} - \eta \nabla g(\omega_{c,t})\} - \omega_c^\star\| \\
&\leq \|\omega_{c,t} - \eta \nabla g(\omega_{c,t}) - \omega_c^\star\| \\
&\leq \|\omega_{c,t} - \eta \nabla F(\omega_{c,t}) - \omega_c^\star\| + \eta \|g(\omega_{c,t}) - \nabla F(\omega_{c,t})\| \\
&\leq \|\omega_{c,t} - \eta \nabla F(\omega_{c,t}) - \omega_c^\star\| + \eta \Lambda
\end{aligned}
$$

Now, we bound $\|\omega_{c,t} - \eta \nabla F(\omega_{c,t}) - \omega_c^\star\|^2$ using $\mu$-strong convexity and $L$-smoothness of $F_c$. The analysis is similar to the convergence analysis in appendix B.1. Thus, for $\eta \leq \frac{1}{L}$

$$
\|\omega_{c,t} - \eta \nabla F(\omega_{c,t}) - \omega_c^\star\|^2 \leq (1 - \eta\mu)\|\omega_{c,t} - \omega_c^\star\|^2
$$

Using this bound we can analyze the original term with $\|\omega_{c,t+1} - \omega_c^\star\|$.

$$
\begin{aligned}
\|\omega_{c,t+1} - \omega_c^\star\| &\leq \sqrt{1 - \eta\mu}\|\omega_{c,t} - \omega_c^\star\| + \eta\Lambda \\
\|\omega_{c,T} - \omega_c^\star\| &\leq (1 - \eta\mu)^{T/2}\|\omega_{c,0} - \omega_c^\star\| + \eta\Lambda\left(\sum_{t=0}^{T-1}(1 - \eta\mu)^{t/2}\right) \\
&\leq (1 - \kappa^{-1})^{T/2}\|\omega_{c,0} - \omega_c^\star\| + \eta\Lambda\left(\sum_{t=0}^{\infty}\left(1 - \frac{\eta\mu}{2}\right)^t\right) \\
&\leq (1 - \kappa^{-1})^{T/2}D + \frac{2}{\mu}\Lambda
\end{aligned}
$$

For the second inequality, we use $\kappa = \frac{L}{\mu}$ and unroll the recursion for $T$ steps. For the third inequality, we use $\sqrt{1-x} \leq 1 - \frac{x}{2}$ and upper bound the finite geometric sum by its infinite counterpart. Finally we use the boundedness of $\mathcal{W}$ and the sum of the geometric series to get our result.

### C.8 Proof of lemma B.2

We present the proof for this lemma here as it is a corollary of lemma C.5.

We utilize the intermediate lemma C.7. Now, if we set $\alpha = \beta = 0$ and $S_c = 1$, we obtain the generalization guarantee for GD on a single node $i \in [m]$. Further, we do not need the terms of $s$ as they appear with $\beta$, and thus, we can choose $s$ very large, so that we can ignore its contribution to error probability. The remainder of the proof follows that of lemma C.5.

### C.9 Proof of lemma C.6

Since $F_c$ is $\hat{L}$-Lipshchitz and $\mu$-strongly convex with minima $\omega_c^\star$,

$$
\begin{aligned}
F_c(w_i^\star) - F_c(\omega_c^\star) &= \frac{F_i(w_i^\star) - F_i(\omega_c^\star)}{Q_c} + \sum_{j \neq i, \mathcal{C}_0(j)=c} \frac{F_j(w_i^\star) - F_j(\omega_c^\star)}{Q_c} \\
&\leq \frac{F_i(w_i^\star) - F_i(\omega_c^\star)}{Q_c} + \sum_{j \neq i, \mathcal{C}_0(j)=c} \frac{F_j(w_i^\star) - F_j(w_j^\star)}{Q_c} \\
&\leq -\frac{\mu \|w_i^\star - \omega_c^\star\|^2}{2Q_c} + \sum_{j \neq i, \mathcal{C}_0(j)=c} \frac{\hat{L}\|w_i^\star - w_j^\star\|}{Q_c} \\
\frac{\mu}{2}\|w_i^\star - \omega_c^\star\|^2 &\leq -\frac{\mu\|w_i^\star - \omega_c^\star\|^2}{2Q_c} + \frac{(Q_c-1)\hat{L}\epsilon_1}{Q_c} \\
\frac{\mu}{2}\|w_i^\star - \omega_c^\star\|^2 &\leq -\frac{\mu\|w_i^\star - \omega_c^\star\|^2}{2Q_c} + \frac{(Q_c-1)\hat{L}\epsilon_1}{Q_c} \\
\|w_i^\star - \omega_c^\star\|^2 &\leq \frac{2\hat{L}\epsilon_1}{\mu} \\
\|w_i^\star - \omega_c^\star\| &\leq \sqrt{\frac{2\hat{L}\epsilon_1}{\mu}}
\end{aligned}
$$

For the first equation, we expand $F_c$ into its component terms, where $Q_c$ denotes the number of correct nodes in cluster $c$. For the second inequality, we use the fact that $w_j^\star = \operatorname{argmin}_{w \in \mathcal{W}} F_j(w)$. For the third inequality, we use strong-convexity of $F_i$ and $\hat{L}$-Lipschitzness for $F_j, j \neq i$. For the fourth inequality, we use a lower bound on $F_c(w_i^\star) - F_c(\omega_c^\star)$ using $\mu$-strong convexity of $F_c$. Finally, we manipulate the remaining terms to obtain the final bound.

## D  Proof of theorem 4.14

By theorem 4.8, $\mathcal{C}_R \neq \mathcal{C}^\star$, with probability $\left(\frac{\rho_2}{m^{(1-\rho_1)}}p\right)^R$. For the $(R+1)^{th}$ step, we bound probability of error by 1. Therefore, with probability $1 - \exp(-\frac{5}{8}R)p$. For the $(R+1)^{th}$ step, we optimize the cluster iterates from `TrimmedMeanGD()` to improve convergence instead of clustering error. Since $\mathcal{C}_{R+1} = \mathcal{C}_R$, each cluster in $\mathcal{C}_{R+1}$ maps to some cluster in $\mathcal{C}^\star$. Without loss of generality, assume that cluster $c \in \mathrm{rg}(\mathcal{C}_{R+1})$ maps to the same cluster $c \in \mathcal{C}$. Now, if $\{c_1, c_2, ..., c_l\}$ are the clusters in $\mathcal{C}_R$ which merged to form cluster $c \in \mathrm{rg}(\mathcal{C}_{R+1})$. Then, we can write

$$
\begin{aligned}
\|\omega_{c,T} - \omega_c^\star\| &= \left\| \frac{1}{l} \sum_{j=1}^{l} (\omega_{c_j,T} - \omega_c^\star) \right\| \\
&\leq \frac{1}{l} \sum_{j=1}^{l} \|\omega_{c_j,T} - \omega_c^\star\| \\
&\leq \frac{1}{l} \sum_{j=1}^{l} \left( \left\| \omega_{c_j,T} - \omega_{c_j}^\star \right\| + \left\| \omega_{c_j}^\star - \omega_c^\star \right\| \right)
\end{aligned}
$$

For the first inequality, we used the definition of $\omega_{c,T}$ from `MERGE()`. For the second inequality, we used the triangle inequality for the $l$ elements. The third inequality is obtained by using triangle inequality and adding and subtracting $\omega_{c_j}^\star$ as defined in appendix C.1.

Now, consider the set of nodes $\{i_1,i_2,...,i_l\} \subseteq [m]$, such that $i_j \in c_j \forall j \in [l]$ and $\mathcal{C}^\star(i_j) = c \forall j \in [l]$. Therefore, we can split each term of $\left\|\omega_{c_j}^\star - \omega_c^\star\right\|$ as –

$$\|\omega_{c,T} - \omega_c^\star\| \leq \frac{1}{l}\sum_{j=1}^{l}\left(\left\|\omega_{c_j,T} - \omega_{c_j}^\star\right\| + \left\|\omega_{c_j}^\star - w_{i_j}\right\| + \left\|w_{i_j} - \omega_c^\star\right\|\right)$$

$$\leq \frac{1}{l}\sum_{j=1}^{l}\left\|\omega_{c_j,T} - \omega_{c_j}^\star\right\| + 2B$$

From lemma C.6, since $i_j$ contributes to both clusters $c_j$ and $c^\star$, we can bound the difference from their minima by $B$. Further, we can use lemma C.5 and the lemma C.7, which is adapted from Theorem 4 in Yin et al. (2018),to bound the convergence of $\left\|\omega_{c_j,T} - \omega_{c_j}^\star\right\|$. If we set $\delta = \frac{1}{nS_{c_j}\hat{L}D}$ and

$$r = \hat{L}\max\{\frac{8d}{nS_{c_j}}\log(1+nS_c\hat{L}D), \sqrt{\frac{8d}{nS_{c_j}}\log(1+nS_c\hat{L}D)}\}$$

$$s = \hat{L}\max\{\frac{4d}{n}(d\log(1+nS_{c_j}\hat{L}D)+\log m), \sqrt{\frac{4d}{n}(d\log(1+nS_{c_j}\hat{L}D)+\log m)}\}$$

where $S_{c_j}$ is the size of cluster $c_j$, we obtain

$$\|\omega_{c,T} - \omega_c^\star\| \leq (1-\kappa^{-1})^{T/2}D + \Lambda' + 2B$$

where

$$\Lambda' = \mathcal{O}\left(\frac{\hat{L}d}{1-2\beta}\left(\frac{\beta}{\sqrt{n}} + \frac{1}{\sqrt{n}c_{\min}}\right)\sqrt{\log(n\max_{j\in[l]}S_{c_j}\hat{L}D)}\right)$$

We can further upper bound $\max_{j\in[l]}S_{c_j}$ by $m$. Now, the probability of error for each cluster $c \in \mathrm{rg}(\mathcal{C}_R)$ for given values of $r$ and $s$ is $\frac{4d}{(1+nc_{\min}\hat{L}D)^d}$, therefore, we can use union bound and multiply this probability of error by $\mathrm{rg}(\mathcal{C}_R) \leq \frac{m}{t}$. Since $t = \Theta(c_{\min})$, we can upper bound this by $\frac{mu''}{c_{\min}}$ for some positive constant $c_{\min}$.

# E  Additional Definitions and Lemmas

We start with reviewing the standard definitions of strongly convex and smooth functions $f: \mathbb{R}^d \mapsto \mathbb{R}$.

**Definition E.1.** $f$ is $\mu$-strongly convex if $\forall w,w'$, $f(w') \geq f(w) + \langle \nabla f(w), w'-w \rangle + \frac{\mu}{2}\|w'-w\|^2$.

**Definition E.2.** $f$ is $L$-smooth if $\forall w,w'$, $\|\nabla f(w) - \nabla f(w')\| \leq L\|w-w'\|$.

**Definition E.3.** $f$ is $L_k$ Lipschitz for every coordinate $k \in [d]$ if, $|\partial_k f(w)| \leq L_k$, where $\partial_k f(w)$ denotes the $k$-th coordinate of $\nabla f(w)$.

**Lemma E.4.** *If $f,g: \mathbb{R}^d \to \mathbb{R}$ are two $\mu$-strongly convex functions on a domain $\mathcal{W}$. Then, $\frac{f+g}{2}$ is also $\mu$-strongly convex on the same domain.*

*Proof.* If $f$ and $g$ are $\mu$-strongly convex on a domain $\mathcal{W}$, then for any $w_1, w_0 \in \mathcal{W}$

$$f(w_1) \geq f(w_0) + \langle \nabla f(w_0), w_1-w_0 \rangle + \frac{\mu}{2}\|w_1-w_0\|^2$$

$$g(w_1) \geq g(w_0) + \langle \nabla g(w_0), w_1-w_0 \rangle + \frac{\mu}{2}\|w_1-w_0\|^2$$

Adding the above equations, we get

$$\frac{f(w_1)+g(w_1)}{2} \geq \frac{f(w_0)+g(w_0)}{2} + \left\langle \frac{\nabla f(w_0)+\nabla g(w_0)}{2}, w_1-w_0 \right\rangle + \frac{\mu}{2}\|w_1-w_0\|^2$$

Thus, $\frac{f+g}{2}$ is also $\mu$-strongly convex. $\square$

**Lemma E.5.** *If $f,g:\mathbb{R}^d\to\mathbb{R}$ are two L-smooth functions on a domain $\mathcal{W}$. Then, $\frac{f+g}{2}$ is also L-smooth on the same domain.*

**Corollary E.6.** *If $f,g:\mathbb{R}^d\to\mathbb{R}$ are two L-Lipschitz functions on a domain $\mathcal{W}$. Then, $\frac{f+g}{2}$ is also L-Lipschitz on the same domain.*

*Proof.* Consider the following term for any $w_1,w_0\in\mathcal{W}$

$$\left\|\frac{\nabla f(w_1)+\nabla g(w_1)}{2}-\frac{\nabla f(w_0)+\nabla g(w_0)}{2}\right\|$$
$$\leq\frac{1}{2}\|(\nabla f(w_1)-\nabla f(w_0))+(\nabla g(w_1)-\nabla g(w_0))\|$$
$$\leq\frac{1}{2}(\|\nabla f(w_1)-\nabla f(w_0)\|+\|\nabla g(w_1)-\nabla g(w_0)\|)$$
$$\leq\frac{1}{2}(L\|w_1-w_0\|+L\|w_1-w_0\|)$$
$$\leq L\|w_1-w_0\|$$

In the second inequality, we use the triangle inequality of norms. For the third inequality, we use the $L$-smoothness of $f$ and $g$. Thus, $\frac{f+g}{2}$ is also $L$-smooth The proof of the corollary is same as above, by replacing terms of $\nabla f$ and $\nabla g$ by $f$ and $g$ respectively. $\square$

**Lemma E.7.** *If each coordinate of a function $f:\mathbb{R}^d\to\mathbb{R}$ is $L_k$-Lipschitz for $k\in[d]$ on the domain $\mathcal{W}$, then $f$ is $\hat{L}=\sqrt{\sum_{k=1}^d L_k^2}$-Lipschitz on the same domain $\mathcal{W}$.*

*Proof.* Consider $w_1,w_0\in\mathcal{W}$.Define a sequence of variables

$\{w[k]=((w_1)_1,(w_1)_2...,(w_1)_k,(w_0)_{k+1},...(w_0)_d)^\intercal\}_{k=0}^d$. Then, $w_1=w[d]$ and $w_0=w[0]$

$$|f(w_1)-f(w_0)|=\left|\sum_{k=1}^d(f(w[k])-f(w[k-1]))\right|$$
$$=\sum_{k=1}^d L_k|(w_1)_k-(w_0)_k|$$

The second inequality follows by using triangle rule. Then, $f(w[k])$ and $f(w[k-1])$ differ only in the $k^{th}$ coordinate, so we apply $L_k$ coordinate-wise Lipschitzness. Now, consider a random variable $v\in\mathbb{R}^d$ such that $v_k=L_k\frac{|(w_1)_k-(w_0)_k|}{(w_1)_k-(w_0)_k}$ if $(w_1)_k-(w_0)_k\neq 0$, else 0. Then,

$$\sum_{k=1}^d L_k|(w_1)_k-(w_0)_k|=\langle v,w_1-w_0\rangle$$
$$\leq\|v\|\|w_1-w_0\|$$
$$\leq\sqrt{\sum_{k=1}^d L_k^2}\|w_1-w_0\|$$

Here, we use the Cauchy-Schwartz inequality for the second step. Then, note that each coordinate of $v$ is bounded by $L_k$. $\square$

