# OpenReview forum: "An Improved Federated Clustering Algorithm with Model-based Clustering"
_TMLR — Accepted by TMLR_

### Review · Reviewer_Xk4R · 2023-12-02

**Summary Of Contributions:**

The paper proposes a novel framework for clustered federated learning called *Successive Refine Federated Clustering Algorithm* to deal with data heterogeneity issue.
1. The algorithm starts with a ONE_SHOT subroutine which trains a local model on each client independently for few iterations and then clusters them based on pair-wise distances.
2. Now under REFINE subroutine, the algorithm runs robust federated learning (TrimmedMean) for each cluster.
3. RECLUSTER subroutine refines the clusters by reallocating the cluster labels for each client based on the distance been local client model and the cluster center model. This aims to reduce the impurity level of each cluster.
4. MERGE subroutine merges the clusters whose pairwise cluster centers distance is less than a set threshold.

The paper presents convergence guarantees of SR-FCA when the loss is strongly convex and smooth, and dist(.,.) is $L_2$ norm. The paper shows that the misclustering error in the first stage of SR-FCA is O(mdexp(−n/√d) ( Lemma 4.6) and the successive stages further reduce the misclustering error by a factor of O(1/m) (Theorem 4.8).

The experimental results are shown on both simulated and real heterogeneous distributed datasets. SR-FCA (with cross-cluster loss) is compared with multiple relevant baselines. On simulated datasets, SR-FCA obtains test accuracy no worse than the best baseline and is able to recover the correct clustering. For CIFAR10, in particular, SR-FCA has 5% better test accuracy than IFCA. On real datasets, SR-FCA outperforms all baselines.

**Audience:**

Yes

**Claims And Evidence:**

Yes

**Requested Changes:**

1. Algorithms 1-4 are hard to follow. Rewrite the algorithms for better understanding and clearly differentiate the worker's computation from the server. Include the participation rate into algorithm 3.
2. An ablation study on $\lambda$ would be interesting.
3. Add more discussion on computation and communication complexity of SR-FCA. This could help in better understanding of the benefits and limitations of the proposed method.

**Strengths And Weaknesses:**

Strengths:
1. The paper proposes a novel clustered federated learning algorithm SR-FCA which doesn't require any specific initialization.
2. The convergence guarantees for the proposed method are presented.
3. The experiments are conducted on both simulated and real heterogeneous distributed datasets.
4. All the relevant baselines are provided and the proposed algorithm either outperforms or is comparable with the baselines.
5. The paper is well written and structured. However, there is room for improvement in Algorithm section (sec 3) and conclusion.
6. It is appreciated that the authors have reported the total time spent for conducting all the experiments and hardware details.

Weaknesses/questions:
1. It is not clear to me why recluster subroutine needs to be before merge. Recluster doesn't effect the model parameters of cluster centers $w_{c,T}$ and merging only changes $w_{c,T}$. Is there any difference in running recluster module before merge and after.
2. In the one_short method: do we have to train the model on each client till convergence? Do all the clients' models have to be initialized to the same point?
3. For the experimental results, can you provide some insights on the accuracy improvements obtained by using TrimmedMean over FedAvg?
4. " Note that the complexity of the REFINE step is the same as that of IFCA in terms of both computation time and communication since in each case, we need to find the loss of every cluster model on every client’s data." It is not clear to me why this statement is true.
For IFCA on the workers end, the compute overhead as compared to traditional FL is because of the cluster estimation which requires $kT$ additional forward passes (referring to line 6 of algorithm 2 in [1]) and the communication overhead is $k \times$ where $k$ is the number of clusters. Can you clarify why it is the same for SR-FCA's refine step?
5. How do we calculate empirical cluster loss $f_c$?
6. Do all the clients have same test dataset?

[1] Ghosh, Avishek, et al. "An efficient framework for clustered federated learning." Advances in Neural Information Processing Systems 33 (2020): 19586-19597.

---

> ### Comment · Reviewer_Xk4R · 2024-01-28
> **Reply to the rebuttal**
>
> I read the updated/rebutted paper and I have a couple more questions/suggestions for the authors to improve the paper.
>
> 1. The pseudo-code is hard to follow. It would be useful if the algorithm contains two parts - server side and client side and then calls the subroutines inside these parts. This makes it easier to understand the communication between the server and the client.
> 2. The cross-cluster loss needs more explanation. In an FL setup, it is important to know who needs what access. For example, to compute the cluster loss for a given client $j$ ie., $f_c(w_j)$, we need access to the data of the entire cluster which is not possible. So how is cluster loss computed?
> 3. It is still not clear why the recluster subroutine needs to be before merge. Recluster doesn't affect the model parameters of cluster centers and merging only changes. Is there any difference between running the recluster module before merge and after?
> 4. For the "Global" (one cluster) baseline, FedAvg has been used. But I think a better global baseline would be Scaffold or FedProx which are meant for heterogeneous data.

---

> > ### Author Response · Authors · 2024-01-31
> > **Response to further comments**
> >
> > Thanks a lot for these helpful comments/suggestions. We address them below.
> >
> > 1. In the revised version that we have uploaded, the pseudocode is already separated into two parts: the client side and server side. We have noted the changes from the original submission in blue, and also indicated this here: https://openreview.net/forum?id=1ZGA5mSkoB&noteId=7J2jPaudLc
> >
> > 2. This is a good point, and we will clarify this further in Section 5 where this metric is defined and discussed. To answer the question: Note that no data is shared for computing cross-cluster loss, rather the models are shared between clients via the server. To compute cluster loss for client $j$ on cluster $c$, $f_c(w_j)$, the server receives the model $w_j$ from client $j$, and sends it to all the clients which belong to cluster $c$. These clients send their empirical losses on model $w_j$ back to the server. The average of these empirical losses is the value $f_c(w_j)$.
> >
> > 3. As we explained in the response above (https://openreview.net/forum?id=1ZGA5mSkoB&noteId=VYUtFN41M2), the ordering of recluster and merge does not change the theoretical analysis at all, and the parts can be thought of as independent. As reported above, we have also done experiments switching their order to get similar empirical performance.
> >
> > 4. We have noted this suggestion. One relevant point: SR-FCA can be used as a meta algorithm that uses TrimmedMean+"any federated algorithm" in the Refine step. Note that, the said federated algorithm can be any of FedProx or Scaffold as well. We will also make this point clear in Section 5.

---

### Review · Reviewer_7Gsg · 2023-12-20

**Summary Of Contributions:**

This paper targets the task of Federated clustering. The main insight of the proposed algorithm is to cluster clients according to the simlarity of the model parameters, and thus striking a balance between handle heterogeneous clients and simultaneous training. Different from the previous proposed approaches, this framework do not need to pre-define the number of clusters. Thorough theoretical analysis are also provided for the proposed framework.

**Audience:**

Yes

**Broader Impact Concerns:**

Broader impact statemnent is not needed in my opinion.

**Claims And Evidence:**

Yes

**Requested Changes:**

1. I suggest the author discuss the communication complexity of this framework and previous methods, as well as potential ways to improve this complexity.
2. I suggest the author include more results on label heterogeneity.

**Strengths And Weaknesses:**

Strength:
1) The proposed similarity metric across different clients (models) is interesting.
2) Extensive theoretical results are provided for the proposed framework;
3) The effectiveness of the proposed method is validated empirically;

Weakness:
1) According to my understanding, the proposed method would require $O(n^2)$ communication complexity to measure the similarity between all pairs of clients (models). I wonder if this can be improved via some heuristic algorithm.
2) The experiments of the paper are mainly focus on domain heterogeneity (e.g., rotation, inversion, etc.). The empirical evidence can be enhanced with more results on label heterogeneity (e.g., Dirichlet heterogeneity as in [1-2]).

[1] Mikhail Yurochkin, Mayank Agarwal, Soumya Ghosh, Kristjan Greenewald, Nghia Hoang, and
Yasaman Khazaeni. Bayesian nonparametric federated learning of neural networks. In International Conference on Machine Learning, pp. 7252–7261. PMLR, 2019.

[2] Hongyi Wang, Mikhail Yurochkin, Yuekai Sun, Dimitris Papailiopoulos, and Yasaman Khazaeni.
Federated learning with matched averaging. arXiv preprint arXiv:2002.06440, 2020a.

---

> ### Author Response · Authors · 2024-01-17
> **Response - 1/2**
>
> We thank the reviewer for their response.
> - **W1:** Note that if the number of clusters, or an appropriate initialization is  not known beforehand, any clustering algorithm, and in turn every federated clustering algorithm, needs to compute the distance between all pairs of clients. Further, note that our distance metric does not much structure which can be utilized for efficient computation, as opposed to say the euclidean distance.
> This procedure is required only once during the initialization, ONE\_SHOT.
>
>     This complexity can be improved by certain heuristic methods, however, our theoretical guarantees would not hold for these procedures, and they might not recover the correct clustering. We propose two such heuristic methods. Both these methods modify the ONE\_SHOT procedure.
>
>     For the first method, we can perform the original ONE\_SHOT procedure on a small randomly sampled set of clients. This would give us some initial clustering for the sampled clients. For the clients which have not been sampled, we can assign them to their closest clusters based on client-cluster distances, similar to RECLUSTER. For the clusters of sampled clients, this imply using a cluster model for the distance metric. As this is unavailable before the REFINE procedure has started, one can use average of the client models in the cluster as the cluster model. If the number of sampled clients is small, around $\mathcal{O}(\sqrt{m})$, then this procedure has time complexity $\mathcal{O}(m)$, where $m$ is the number of clients.
>
>     For the second method, we modify the graph generated by peforming all pairwise comparisons by using a weighted graph.  Instead of computing all edges, i.e., all pairwise distances, we can compute only a fraction of them. For instance, for each client, we can sample a random subset of remaining clients as pairs whose distances we compute. After comparison, we either put an edge between two clients (edge weight $1$) or no edge between them (edge weight $0$). For all pairs, whose distances have not been computed, we can use either an edge weight of $0.5$, to denote the uncertainty. We can then perform correlation clustering on this graph to obtain clusters. Note that the time complexity for this method is also $\mathcal{O}(m)$, if only a constant number of edge weights are computed per client.
>
>
> - **W2:** FEMNIST and Shakespeare, the real federated datasets from LEAF, have high label heterogeneity among the clients. Therefore, our experiments on these datasets, provided in Table 3 of our manuscript, should provide evidence for the performance of our method in the presence of label heterogeneity.

---

> > ### Author Response · Authors · 2024-01-18
> > **Response - 2/2**
> >
> > The requested changes are addressed below.
> >
> > 1.  The main overhead of SR-FCA is during ONE\_SHOT, where we need to compute pairwise distances, requiring $\mathcal{O}(m^2)$  forward passes. The REFINE step is comparable to IFCA and we provide a detailed analysis below.
> > Assume that we run  IFCA on $C$ clusters for $T$ rounds with $E$ local steps with $\alpha$ fraction of $m$ clients participating in each round. Then, we require $\mathcal{O}(\alpha m TE)$ backward passes and $\mathcal{O}(\alpha m T)$ communication for local training and aggregation.  For cluster identity estimation, in each round, we need to compute the loss of each cluster model on every participating client. To send cluster models to each client, we need $\mathcal{O}(\alpha m C T)$ communication and to compute losses, we need $\mathcal{O}(\alpha m C T)$ forward passes.
> >
> >
> >
> >     If we run $R$ REFINE  steps with $\leq C$ clusters per step where each  TrimmedMeanGD procedure runs for $T$ rounds with $E$ local steps and $\alpha$ fraction of clients participating in each round. We need $\mathcal{O}(\alpha m E T R)$ backward passes and $\mathcal{O}(\alpha m T R)$ communication for aggregation and local training. For clustering in  REFINE and  MERGE steps, we need to compute the loss of every model on every client, which requires $\mathcal{O}(m C R)$ communication and $\mathcal{O}(m C R)$ forward passes.
> >
> >     We summarize these results in the following table.
> >
> >     | Algorithm | Communication | Training Steps |  Forward Passes |
> >     |---|---|---|---|
> >     |     IFCA | $\mathcal{O}(\alpha m TC)$ | $\mathcal{O}(\alpha m T E)$ | $\mathcal{O}(\alpha m T C)$ |
> >     |     SR-FCA | $\mathcal{O}(\alpha m T R + m CR)$ | $\mathcal{O}(\alpha m TER)$ | $\mathcal{O}(m C R)$|
> >     **Table 1** Communication, training runtime and forward passes for loss computation in clustering for  SR-FCA and  IFCA. $E$ is the number of local steps, $T$ is the number of rounds, $R$ is the number of  REFINE steps, $C$ is the number of clusters and $\alpha$ is the fraction of clients participating per round.
> >
> >
> >     Comparing the two algorithms, where the number of  REFINE steps $R$ and the number of clusters $C$ are assumed to be constants, for a constant $\alpha$,  both algorithms require the same communication and backward passes, but IFCA  needs more forward passes for clustering. If $\alpha = \Theta(\frac{1}{m})$, which corresponds to selecting a constant number of clients per round, then if $m = \Omega(T)$,  IFCA is better than SR-FCA  in terms of communication and forward passes, and vice-versa if $T=\Omega(m)$.
> >
> >
> >
> >
> >     As for the other baselines, Local-KMeans is the most efficient in terms of computation and communication complexity as it performs only a single round of communication, at the end of training. CFL, on the other hand, adopts a top-down approach to split clusters into two parts based on the cosine similarity of gradients. Each split calls a bipartitioning subroutine with runtime $\mathcal{O}(m^3)$ making it the slowest baseline.
> >
> >
> >
> >     We will add the discussion of computation and communication complexity  from this rebuttal to the main draft, after all reviews have been uploaded.
> >
> >
> >
> > 2. We have performed experiments on real federated datasets, FEMNIST and Shakespeare, which have high label heterogeneity.

---

> ### Comment · Reviewer_7Gsg · 2024-02-06
> **Final recommendation**
>
> The reviewer's concern raised in previous review process are properly addressed. Specifically, it is now clear to the reviewer and readers that the communication cost of the proposed algorithm does not have to reach $O(m^2)$. Although there are still some remaining concerns on the empirical validation (e.g., experiments on label heterogeneity with larger dataset such as CIFAR10 should be included), the reviewer recommands acceptance on this paper given the interesting framework proposed.

---

### Review · Reviewer_RdDU · 2024-01-05

**Summary Of Contributions:**

The paper introduces a "novel" clustering framework for training local models in a federated learning setup. The key contribution of the work is an iterative refinement algorithm (reminiscent of the EM-algorithm), its theoretical and experimental analysis. The key idea is to cluster "similar clients" in the federated setup  and train individual models for each cluster. The aim here is to address heterogeneity in client data. Similarity of clients is defined in terms of distance between the locally trained model weights on the client data. Prior works address the heterogeneity by training local models for each client. The key advantage here seems to be that they are able to leverage data across a few similar clients.

**Audience:**

Yes

**Claims And Evidence:**

Yes

**Requested Changes:**

As mentioned in "Strengths and Weaknesses", the following are the requested changes:

1. Fix the terminology issues, ensure that the terms such as C_0, T, rg(.) etc are appropriately defined before usage.

2. The experimental section needs to be expanded significantly. Firstly, the reproducibility issue must be fixed: give a detailed description of the all the parameters, hyper parameter ranges used. Include an empirical analysis of the algorithm runtime. More detailed experiments are requested analysis the impact of a) size of the model used b) change in the participation fraction of the clients c) number of clients (which seems to be too high for the 2-4 clusters used) and data size with each client. I think the datasets are either synthetic or semi-real (because the rotations based clusters are created). It would be great if some experiments can be included involving real heterogeneity (this needs some thinking through but perhaps images from different classes - cats vs dogs, could be one starting point).

**Strengths And Weaknesses:**

Strengths:

1. Introduction of a reasonably intuitive framework.
2. Theoretical analysis is comprehensive, albeit a bit derivative of prior works.

Weaknesses:

1. The paper's notation is all over the place. For example, on page 4, the usage of terms such as C_0 precedes before defining them (which they do later in the page 5 in the description of the routines). T (in w_{I,T}) is undefined similarly until page 5. Further terms like rg{C} are never defined and it took the reader a while to figure out what they meant.
2. Apart from the framework, I feel the novelty is limited. Indeed the algorithm is a natural variant of the standard algorithm used in k-means. The analysis needs some work but uses standard apparatus (union bounds and probabilistic methods).
3. The biggest let down, in my opinion, is the experimental analysis. It needs to be lot more comprehensive. Firstly, a lot of detail is missing from a reproducibility point of view: for example, the size of the neural network model used is not specified. Indeed this is important, as the similarity metric is defined based on the weight vector. Also, it is mentioned that only a fraction of clients participate in the model training. What fraction is used? If the fraction is too low, it could explain why the performance of the models on a simple dataset such as MNIST is so low (<90%) for the global model. Also, it would be instructive to see what the impact of model size would be on the performance of the algorithm w.r.t the baselines. Indeed, the whole setup seems to create an unnatural performance gradient from the baseline. For instance, simply taking a convolutional neural network, instead of feedforward network, could boost the model performance across the board and diminish the impact of the proposed algorithm. The best known models on MNIST yield a >99% accuracy.
4. One of the key claims that the paper makes is that it gets rid of the initialization problem. However, since they need to perform a rather expensive hyper parameter tuning on the $\lambda$, this seems like a moot point. Indeed, one of the aspects missing from the results is the runtime analysis for different methods (including HPO).

---

> ### Author Response · Authors · 2024-01-18
> **Response - 1/2**
>
> We thank the reviewer for their response.
>
> - **W1:** Apologies for the confusion with the notations. Note that the terms $ \mathcal{C}$ and $ w_{i,T} $ have been introduced in Algorithms 1-4, while the $ rg $ notation was introduced in the 3rd paragraph on page 4. We use these notations while describing the algorithm in Section 3 (pages 4-6) while the actual algorithm is on page 5, which may have led to the confusion. We have updated the draft to include a definition of these terms before they have been used.
>
> - **W2:** We are a bit confused by this comment. We assume the reviewer means the Lloyd's algorithm by the standard k-means. We actually do not use Lloyd's in SR-FCA; instead, we use a simple distance-based thresholding to declare whether clients are in same cluster or not. Only the recluster step has some similarity with standard k-means where we do a reassignment of clients to cluster - but then the merge step is also different from k-means. We do indeed use probabilistic methods - but that term is very broad.
>
> - **W3:**
>     - In the first two paragraphs of section 5, we do mention the models used for each dataset: 2-layer feedforward NN for MNIST, ResNet9 for CIFAR10, a CNN for FEMNIST and 2-layer stacked LSTM for Shakespeare. While we do not mention all the details about these models, for instance the number of hidden units in each layer of the model or the learning rate used,  we have provided the code in the supplementary material, which contains all these details and can be used to reproduce all our experimental results. We are providing the number of units used in the models:
>         1. MNIST: 2 fully connected networks have 2048 hidden units and 10 units with a ReLU activation,
>         2. CIFAR10 : The layer-wise description of ResNet9 can be obtained from [2].
>         3. FEMNIST : The CNN has 2 sets of 2D convolution and max pooling layers. Convolution layers have $(5,5)$ filter size, containing $32$ and $64$ filters respectively, and are followed by a ReLU activation. Pooling layers have filter size $(2,2)$ and stride length $2$. The output of the convolution layers is fed to 2 fully connected layers of $2048$ and $10$ units respectively with ReLU activation.
>         4. Shakespeare : The stacked LSTM takes as input sequences of  $80$ character. These are passed through an embedding layer of hidden dimension $8$. These embeddings are passed into a $2$-layer LSTM with $256$ hidden units. The output of LSTM is fed to a hidden layer with ReLU activation and $80$ units. The hidden states for LSTMs are initialized as vectors with each entry being $0$. For all other details related to reproducibility, we refer to the code provided in supplementary material.
>
>     - Note that the similarity metric depends not just on weight vectors, or the model size, but rather on how well a model trained on one client can perform on other clients.
>     - We do mention the fraction of clients that participate in model training in the first paragraph of page 9. For MNIST, all clients participate in training, but for CIFAR10 only 50\% of the clients participate. For FEMNIST and Shakespeare, we sample $m=50$ clients.
>     - Note that the performance of the global model for the MNIST examples does not reach 99\% even though MNIST is a simple dataset precisely due to the heterogeneity in the dataset. Specifically, it is hard for a single global model to simultaneously learn MNIST dataset where the images might be rotated or inverted. This is also the motivation for clustered FL, as inside a cluster, images might have the same rotation or inversion, thus allowing for better performance. Further, previous attempts at using Rotated MNIST in IFCA[1] and Continual Learning[3] can only obtain test accuracies in the range 80-90\% but not better. This also holds for FEMNIST (which is a version of EMNIST) or CIFAR10. For each dataset, we have specifically used models that obtain best performance in FL literature[4,5],  therefore the gains in performance are due to our clustering algorithm and using a stronger model should not diminish these gains.

---

> > ### Author Response · Authors · 2024-01-18
> > **Response -  2/2**
> >
> > - **W4:** Note that we need to perform hyperparameter tuning for every federated clustering algorithm -- $\lambda$ and $\beta$ for SR-FCA, the number of clusters $K$ for IFCA, and Local-KMeans and parameters $\epsilon_1,\epsilon_2$ and $\gamma_{\max}$ for CFL. Even with the best tuned $K$, IFCA still requires an appropriate initialization to obtain best performance.
> >
> >     The only case where we do not need to tune for the number of clusters is for simulated datasets, where, from Figure 1, the distribution of $\lambda$ has a large gap between clients in same cluster and clients in different cluster which makes tuning $\lambda$ easy.
> >
> >     [**Figure 1**](https://anonymous.4open.science/r/sr-fca-307C/inv_mnist.jpg) :  Pairwise distances and threshold $\lambda$ for Inverted MNIST
> >
> >
> >     Further, for each algorithm, we ran $20$ trials of random search to tune all its hyperparameters, to ensure all algorithms had similar budget for hyperparameter tuning. Note that we did not record the actual time for the full hyperparameter tuning for each case, but we do provide the time for a single run for each algorithm in Table 1. The runtime of SR-FCA is comparable to other clustered FL baselines like IFCA and CFL.
> >
> > | Dataset |SR-FCA | REFINE steps only | Local | Global | IFCA | CFL | Local-KMeans|
> > |---|---|---|---|---|---|---|---|
> > |MNIST Rotated|  80  | 62  | 18  | 10 | 50  | 87  | 23 |
> > |MNIST  Inverted|   18  | 5  | 13  | 5  | 15  | 20  | 17 |
> > **Table 1** : Time (in min) taken to run baselines on given datasets for a single seed
> >   Finally, we emphasize that we have provided the code for reproducing all experiments, including the hyperparameter tuning.
> >
> >
> > **Requested Changes:**
> >
> > - We have updated the draft by defining the terms $\mathcal{C}_0, T$ and $rg$ in both the algorithm and the main text, before they are used. We have also used $K$ to denote the number of clusters in $\mathcal{C}^\star$, instead of $C$ to avoid confusion.
> > - We would like to emphasize that we have included the code to reproduce all the experiments, including the hyperparameter tuning. We have still included a section in the appendix on all the hyperparameters used and their ranges.
> >
> >     As explained in our rebuttal to **W3**, the size of the model used should not impact the algorithm's performance as we have used the appropriate models for each dataset and existing results in literature do not obtain better performance than those reported by us.
> >
> >     The participation fraction is $100\%$ for MNIST datasets and $50\%$ for CIFAR10, which has been mentioned in Section 5, first paragraph. As this is sufficiently high, we are unsure if changing this should change the performance.
> >
> >     We do use real federated datasets with real heterogeneity, FEMNIST and Shakespeare, with their results provided in Table 3 of our draft. These datasets have real label heterogeneity which is an instance of the "cats vs dogs" example suggested by the reviewer.
> >
> > **References**
> > 1. Ghosh et al. An efficient framework for clustered
> > federated learning. NeurIPS 2021.
> > 2. Page. https://myrtle.ai/learn/how-to-train-your-resnet-4-architecture/
> > 3. Lopez-Paz and Marc' Aurelio Ranzato. Gradient episodic memory for continual learning. NeurIPS 2017.
> > 4. Caldas et al. LEAF: A benchmark for federated settings. 2018.
> > 5. Reddi et al. Adaptive Federated Optimization. ICLR 2021.

---

> > > ### Comment · Reviewer_RdDU · 2024-02-07
> > > **Final Recommendation**
> > >
> > > Thank you authors, for the detailed comments (and modifications) on the reviewer comments. They address most of my key concerns and I am voting for acceptance.

---

### Author Response · Authors · 2024-01-18
**Updates in the submission**

We have updated our submission to incorporate the reviewers' suggestions. These updates are highlighted in blue and we summarize them below.
- **Notation:** Based on Reviewer RdDU's suggestions, we have added additional description of the notations used in the algorithm description in Sections 2 and 3. We have also used $K$ to denote the number of clusters in $\mathcal{C}^\star$ instead of $C$ to avoid confusion.
- **Algorithm:** Based on Reviewer Xk4R's suggestions, we have split Algorithms 1 and 3 into Server and Client subroutines and added implementation details in Section 3.
- **Communication and Computation Complexity:** Based on suggestions from Reviewer Xk4R and 7Gsg, we have added a new section ( Section 3.4), where we perform a detailed comparison between SR-FCA and baselines in terms of communication and computation complexity.
- **Pairwise distances and $\lambda$:** Based on suggestions of Reviewer Xk4R and RdDU, we have added a discussion and Figures 3-4 in Section 5.1 to evaluate the distribution of pairwise distances and the choice of $\lambda$.

---

### Decision · Action_Editor_VCRH · 2024-02-08

**Recommendation:** Accept with minor revision

**Comment:**

The paper was reviewed by three reviewers with expertise ranging from theory to more practical aspects of the clustering and federated learning. The reviewers gave extensive initial feedback which was successfully addressed by the authors through revisions and responses. All reviewers are satisfied with the revised paper and recommending the paper to be (conditionally) accepted modulo small minor remaining comments detailed below:

* Please include experiments on label heterogeneity with a larger dataset, such as CIFAR10 to showcase the scalability of the proposed framework.

* Please revise the paper for more clarity in response to Reviewer Xk4R's latest comments.

Congratulations!

**Audience:**

The paper is well within the scope of TMLR.

**Claims And Evidence:**

The paper provides a clustering framework in the context of federated learning that provably does not need any warm start. The claims of the paper are both theoretically and empirically validated.

---

> ### Author Response · Authors · 2024-02-26
> **Minor Revisions**
>
> We have incorporated the suggested minor revisions in our updated camera-ready version.
>
> - **CIFAR10 with Label heterogeneity**: We create 2 clusters where the clients in the first cluster have the first $7$ labels while those in the second cluster have the last $7$ labels. We report the results for all baselines on this dataset, referred to as Label CIFAR10 in Tables 2, 3, and 5. Similar to Rotated CIFAR-10, SR-FCA recovers the correct clustering and comprehensively outperforms all baselines.
> - **Reviewer Xk4R's suggestions**: We have updated the paper to incorporate Reviewer Xk4R's latest comments. In the first paragraph 1 on page 11, we have described the exact procedure of computing the cross-cluster distance metric, and in the third paragraph on page 11, we have clarified why using FedAvg instead of SCAFFOLD or FedProx is justified.